# Aligning Language Models with Human Preferences via a Bayesian Approach

**Jiashuo WANG**[1]**, Haozhao WANG**[2]**, Shichao SUN**[1]**, Wenjie LI**[1*]
[1]Department of Computing, The Hong Kong Polytechnic University
[2]School of Computer Science and Technology, Huazhong University of Science and Technology
{csjwang,csssun,cswjli}@comp.polyu.edu.hk
hz_wang@hust.edu.cn

## Abstract

In the quest to advance human-centric natural language generation (NLG) systems, ensuring alignment between NLG models and human preferences is crucial. For this alignment, current popular methods leverage a reinforcement learning (RL) approach with a reward model trained on feedback from humans. However, inherent disagreements due to the subjective nature of human preferences pose a significant challenge for training the reward model, resulting in a deterioration of the NLG performance. To tackle this issue, previous approaches typically rely on majority voting or averaging to consolidate multiple inconsistent preferences into a merged one. Although straightforward to understand and execute, such methods suffer from an inability to capture the nuanced degrees of disaggregation among humans and may only represent a specialized subset of individuals, thereby lacking the ability to quantitatively disclose the universality of human preferences. To address this challenge, this paper proposes a novel approach, which employs a Bayesian framework to account for the distribution of disagreements among human preferences as training a preference model, and names it as **d-PM**. Besides, considering the RL strategy's inefficient and complex training process over the training efficiency, we further propose utilizing the contrastive learning strategy to train the NLG model with the preference scores derived from the d-PM model. Extensive experiments on two human-centric NLG tasks, i.e., emotional support conversation and integrity "Rule-of-Thumb" generation, show that our method consistently exceeds previous SOTA models in both automatic and human evaluations.

## 1 Introduction

Human-centric natural language processing (NLP) aims to develop NLP systems that are finely attuned to human preferences [14, 12, 30]. Consequently, learning from human feedback is well-suited for training models in human-centric NLG tasks [18, 28]. Currently, reinforcement learning (RL) with a reward model is the most popular method to align models with human preferences [26, 11, 41]. Its effectiveness depends heavily on how well human preferences are learned by the reward model [6, 4]. However, modeling human preferences can be challenging.

Due to the high subjectivity of personal standards and human values, it can be difficult to reach a consensus on preferences among individuals, significantly increasing the learning difficulty. As depicted in Figure 1, persons may have varied preferred responses with inconsistent emotions and values given the same context. To tackle this challenge, existing methods mostly adopt aggregation techniques such as majority voting or averaging [11, 6, 4]. However, aggregated preferences potentially cater

---

[*] Corresponding author.

37th Conference on Neural Information Processing Systems (NeurIPS 2023).

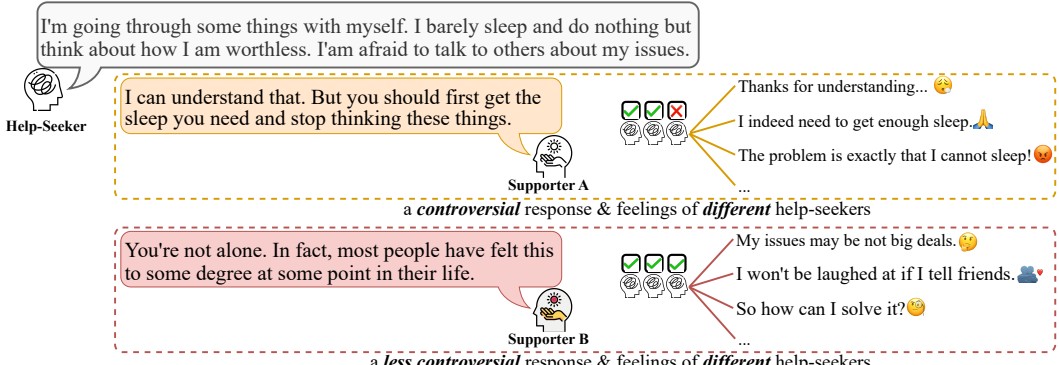

Figure 1: People can have different feelings towards the same response in the emotional support conversation because of their own experiences and values. A trustworthy human-centric system is expected to consider the benefits of universal groups, including minorities, and generate less controversial and more helpful content, like supporter B instead of A.

to specific subsets of people, risking generating controversial content (see Supporter A in Figure 1). Additionally, subjectivity and inconsistency are intrinsic components of certain tasks, such as emotion analysis [13] and ethical evaluation [10], necessitating careful consideration instead of dismissal [3]. Therefore, for widely acceptable and less controversial outputs, it is necessary for the resulting NLG systems to account for capturing disagreements inherent in human preferences [17].

In this paper, we introduce a novel Bayesian-based approach termed Preference Modeling with Disagreement (**d-PM**). This method is designed to approximate a "universal preference" that comprises the preferences of "all individuals", given the preferences of several individuals. Although a soft label, derived from these several individuals, can intuitively account for disagreement, outliers or extreme labels can disproportionately influence the overall perception. Therefore, we employ Bayesian inference to refine these preferences. Specifically, the observed preference among selected individuals serves as prior knowledge. Our d-PM aims to leverage distribution of all possible universal preferences (likelihood probability) to adjust and smooth the initially observed one, leading to the derivation of a universal preference (posterior). Upon obtaining the universal preference, we calculate the likelihood of the expected preference types to establish a preference score, which is then utilized for further language model alignment.

Based on the d-PM model, we further optimize the language models to generate widely acceptable and less controversial texts. Specifically, we propose utilizing the contrastive learning strategy to calibrate the generation model towards generating texts with high preference scores provided by d-PM. Although existing RL strategies can also be leveraged to make the calibration, they are generally perceived as costly in terms of convergence [9] and online decoding processes [40]. We assess our proposed method on two human-centric NLG tasks: emotional support conversations and moral integrity RoT generation. Experimental results demonstrate that our framework can be applied to state-of-the-art generation models for each task without performance degradation and meanwhile effectively increases global consensus of human preferences embedded in generated texts.

Our main contributions are three-fold: **(i)**. To the best of our knowledge, we are the first to align text generation models with human preferences while considering inherent disagreement among different individuals. **(ii)**. In order to model human preferences with their disagreement, we propose a Bayesian approach, Preference Modeling with Disagreement (d-PM). Additionally, we use its preference scores to calibrate NLG models via contrastive learning for generations that can be widely acceptable and less controversial. **(iii)**. We conduct experiments on two human-centric NLG tasks, i.e., emotional support conversations and integrity RoT generation. Experimental results demonstrate the effectiveness and versatility of our proposed method. [2]

---

[2]Our codes are released at `https://github.com/wangjs9/Aligned-dPM`.

## 2 Framework

Aligning text generation models with human preferences requires two essential components: modeling human preferences and calibrating the text generation model.

**Preference Modeling with Disagreement**   Human preferences can be inferred from a human-annotated dataset, denoted as $\mathcal{D}$. Each instance in the dataset is represented as a triplet $(c, s, l)$. Here $c$ is a context; $s$ is a text; and $l$ is a label indicating the annotators' preferences. The inherent disagreement in human preferences can be encapsulated within the label in two distinct ways. In the first approach, the label can be a soft label derived from multiple annotations, all attributed to the same sentence [13]. These annotations are sourced from multiple human annotators to preserve disagreement among individuals. The second approach is the direct collection of global consensus. In this context, the label signifies the proportion of people who find a particular sentence acceptable, an estimate provided by a single human annotator [42, 7].

Aimed at capturing human preference with disagreement within the dataset, we assume there is a distribution $\rho$ over two classes, $\{acceptable, unacceptable\}$, comprising preferences of all humans, and therefore $l$ is the sampling result from $\rho$. We employ a preference model $\mathcal{R}(\theta)$ to infer $\rho$ give $c$ and $s$ as inputs and the probabilistic format of $l$ as the prior distribution. Since we focus on whether $s$ is widely acceptable, the likelihood of the class $acceptable$ is defined as the preference score:

$$\mathcal{S}_{(s,c)} = \mathcal{R}(s, c; \theta)_{acceptable}. \tag{1}$$

**Calibration for Alignment**   In aligning NLG models with human preferences, we calibrate the existing generation model $\mathcal{G}(\xi_0)$ with preference scores. Here, $\mathcal{G}(\xi_0)$ stands for a model already fine-tuned on a dataset $(X, Y)$, where $X$ and $Y$ represent the input set and the corresponding output set, respectively, and $\xi_0$ are the optimized parameters. Significantly, if the dataset for preference modeling is identical to the $(X, Y)$ dataset, then $(x, y) \in (X, Y)$ corresponds to $(c, s) \in \mathcal{D}$. If not, these two datasets should belong to a similar domain. We initially decode $K$ candidate sequences $\{\tilde{y}_k\}_{k=1}^{k=K}$ via $\mathcal{G}(\tilde{y}_k|x; \xi_0)$ for each $x \in X$. Then, we further train $\mathcal{G}(\xi_0)$, aimed at a new objective: aligning the likelihoods of candidate sequences with preference scores $\{\mathcal{S}_{(\tilde{y}_k, x)}\}_{k=1}^{k=K}$.

## 3 Method

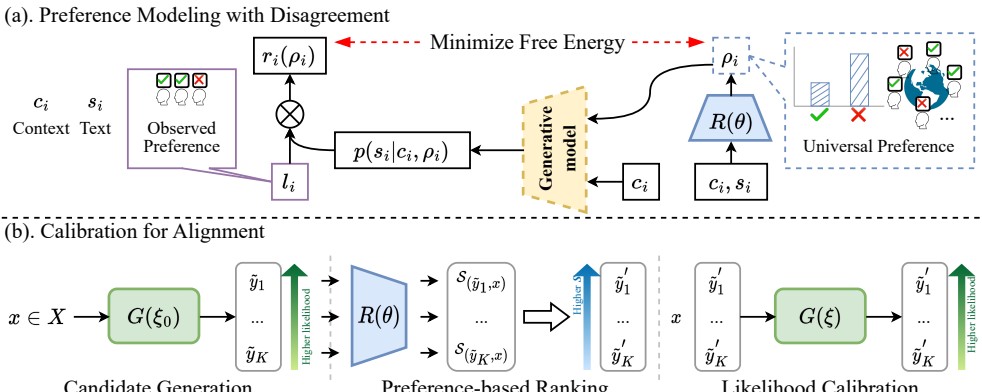

Figure 2: Diagram for preference modeling with disagreement and calibration for alignment.

This section presents our method, whose diagram is shown in Figure 2. We first use a Bayesian approach, i.e., d-PM, to model human preferences with disagreement (Section 3.1). Then we calibrate a text generation model by contrastive learning with preference scores of d-PM to align this model with human preferences (Section 3.2).

## 3.1 Preference Modeling with Disagreement

We establish a distribution $\rho$ to represent the universal preference for the text $s$ given its context $c$. Therefore, the observed annotations $l$ are considered as samples from $\rho$, and can form a prior distribution $p_i(\rho)$. Inspired by [29], we devise a Bayesian approach to approximate $\rho$ using this prior.

Specifically, we establish a connection between $\rho$ and $l$ through the optimization process of a generative model. This model is designed for generating text $s$ conditioned on $c_i$ and $\rho$: $p(s|c_i, \rho)$. The log-likelihood of the text can be formulated as $\sum_i \log p(s_i|c_i) = \sum_i \log \left( \sum_\rho p(s_i|c_i, \rho)p_i(\rho) \right)$, where $p_i(\rho)$ is the prior preference distribution. Its optimization can be achieved by introducing a variational posterior distribution $q(\rho|s_i, c_i)$ for the $i$-th datapoint, and minimizing the free energy (negated evidence lower bound) formulated as:

$$-\sum_i \log p(s_i|c_i) + \sum_i \sum_\rho q(\rho|s_i, c_i) \log \frac{p(s_i|c_i, \rho)p_i(\rho)}{q(\rho|s_i, c_i)}. \tag{2}$$

Minimization of the free energy involves estimations of both the forward distribution of text $s_i$: $p(s_i|c_i, \rho)$, and the posteriors $q(\rho|s_i, c_i)$, which can be computed by our preference model:

$$q(\rho|s_i, c_i) = \mathcal{R}(s_i, c_i|\theta). \tag{3}$$

As for $p(s_i|c_i, \rho)$, it is defined only on the $i$-th datapoint and is computed by minimizing Equation (2) for fixed $q(\rho|s_i, c_i)$, s.t., $\sum_i p(s_i|c_i, \rho) = 1$ for all $\rho$. Thus, the optimum is achieved by:

$$p(s_i|c_i, \rho) = a_{i,\rho} = \frac{q(\rho|s_i, c_i)}{\sum_j q(\rho|s_j, c_j)}. \tag{4}$$

From Equation (4), the generative model can be regarded as a matrix of variables $a_{i,\rho}$ describing conditional probabilities of different responses $s_i$ given different latent distributions $\rho$ with known $c_i$.

In a variational way, Equation (2) can be rewritten as $-\log p(s_i|c_i) + \sum_i \text{KL}(q(\rho|s_i, c_i)\|r_i(\rho))$. Here, $r_i(\rho) \propto p(s_i|c_i, \rho)p_i(\rho)$ is the posterior model of the generative model, and it can be reformulated with reduction of $p(s_i|c_i, \rho)$ to the matrix in Equation (4):

$$r_i(\rho) = \alpha_i \cdot p_i(\rho)p(s_i|c_i, \rho) = \alpha_i \frac{p_i(\rho)q_i(\rho|s_i, c_i)}{\sum_j q_j(\rho|s_j, c_j)}, \tag{5}$$

where $\alpha_i$ is a scalar enabling $\sum_\rho r_i(\rho) = 1$. Accordingly, we can minimize the KL divergence between $q(\rho|s_i, c_i)$ and $r_i(\rho)$ to minimize the free energy. The minimization of the free energy in Equation (2) can be derived as:

$$\sum_i \text{KL}(q(\rho|s_i, c_i)\|r_i(\rho)) = \min_\theta \sum_i \text{KL}\left(\mathcal{R}(s_i, c_i; \theta) \middle\| \alpha_i \cdot p_i(\rho)\frac{\mathcal{R}(s_i, c_i; \theta)}{\sum_j \mathcal{R}(s_j, c_j; \theta)}\right). \tag{6}$$

By optimizing the above objective, we can optimize the parameters of our preference model, i.e., $\theta$.

## 3.2 Calibration for Alignment

It is nearly possible for $\mathcal{G}(\xi_0)$ to generate texts with both high and low preference scores, shown in Figure 3. However, we expect to calibrate the model such that the generation probability aligns with these preference scores. Specifically, we use diverse beam search [34] to generate multiple candidates and then use our d-PM to evaluate these candidates. For the sake of more likely generating a high preference score text, we propose a model-agnostic module to leverage contrastive learning to calibrate generation likelihood aligning with d-PM. Taking inspiration from recent calibration work [32, 23, 39], we implement this module through the following three steps:

**Step 1: Candidates Generation.** We generate candidates from the text generator $\mathcal{G}(\xi_0)$, which has been fine-tuned on corresponding dataset $(X, Y)$ and its parameters are $\xi_0$, on its own training dataset. Given an input sequence $x \in X$, we first use $\mathcal{G}(\xi_0)$ to generate $K$ candidates $\{\tilde{y}_1, \tilde{y}_2, \cdots, \tilde{y}_K\}$ using diverse beam search. As a result, these candidates will get similar possibilities yet different preference scores according to the above preliminary study.

**Step 2: Preference-based Ranking.** We use our proposed d-PM $\mathcal{R}(\theta)$ to measure the preference score $\mathcal{S}_{(\tilde{y}_k, x)}$ of each candidate $\tilde{y}_k$. Then we rank these candidates according to the above preference score and obtain a list of ranked candidates: $\tilde{y}'_1, \tilde{y}'_2, \cdots, \tilde{y}'_K$, where $\mathcal{S}_{(\tilde{y}'_i, x)} > \mathcal{S}_{(\tilde{y}'_j, x)}$ for $\forall\, i < j$.

**Step 3: Likelihood Calibration.** As mentioned before, we leverage contrastive learning to assign higher likelihoods to the candidates with higher preference scores. The following pairwise margin loss is used to adjust the generator $\mathcal{G}(\xi)$.

$$\mathcal{L}^r = \sum_i \sum_{j>i} max(0, \mathbf{P}(\tilde{y}'_j; \xi) - \mathbf{P}(\tilde{y}'_i; \xi) + \lambda_{ij}), \quad (7)$$

where $\lambda_{ij}$ is the default margin $\lambda$ multiplied by the difference in rank between the samples, i.e., $\lambda_{ij} = \lambda * (j - i)$. $\mathbf{P}(\tilde{y}'_i; \xi)$ is the length-normalized log-probability of the candidate:

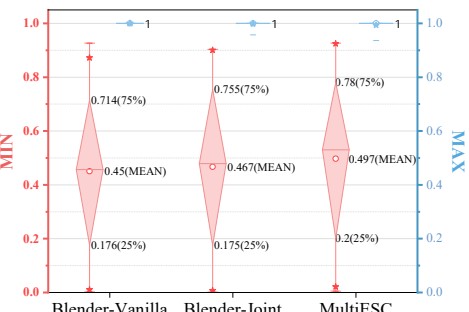

Figure 3: The maximum and minimum preference scores of 10 candidates generated via diverse beam search given the same context. We test on 1000 data instances and three emotional support conversation models.

$$\mathbf{P}(\tilde{y}'; \xi) = \frac{\sum_{t=1}^{|\tilde{y}'|} \log \mathcal{G}(\tilde{y}'_t | x, \tilde{y}'_{<t}; \xi)}{|\tilde{y}'|^\alpha}, \quad (8)$$

where $\alpha$ is the length penalty hyperparameter. To avoid forgetting token-level likelihood information of the ground-truth text, we also use an additional token-level negative log-likelihood. The final calibration loss is as follows:

$$\mathcal{L}^c = -\lambda \frac{1}{|y|} \sum_{t=1}^{|y|} \log \mathcal{G}(y_t | x, y_{<t}; \xi) + \mathcal{L}^r. \quad (9)$$

We minimize $\mathcal{L}^c$ to optimize the generator's parameters $\xi$. This process is supervised by our d-PM model and aligns the generation model with human preference.

## 4 Experiments

### 4.1 Emotional Support Conversation

In an emotional support conversation, a supporter aims to buffer a help-seeker's emotional distress and help the help-seeker to change the difficult situation [22]. In this context, the model functions as the supporter, while the user is always the help-seeker. Due to different personal experiences, different help-seekers may respond with varied feelings and reactions to the same response, as illustrated in Figure 1. Our objective is to enhance the model's ability to generate responses that will not escalate the negative feelings of a diverse range of help-seekers.

**Dataset and Base Models** The benchmark ESConv [22], containing approximately 1k conversations with 31k utterances, develops each conversation between a help-seeker and a supporter. We include BlenderBot-Vanilla and BlenderBot-Joint proposed in conjunction with the dataset, and the SOTA model MultiESC[5]. We reproduced the base models in accordance with their respective papers and publicly available codes.

**Human Preferences with Disagreement** We derive human preferences from the Motivational-Interviewing-Dataset [36]. This dataset encompasses around 17k supporter responses to help-seekers. Each response is annotated by $2 \sim 4$ experts following the MI codes[25]. The labels can be induced into two classes $\{acceptable, unacceptable\}$, and the human preferences with disagreement can be estimated by our d-PM method. By fine-tuning a BERT model on this dataset using prefix-tuning [37, 19], we obtain the d-PM. Additional details can be found in Appendix B.2.

Table 1: Automatic evaluation results on ESConv. All results are significantly better than the corresponding base model with $p < 0.01$.

| Model | | B-1 | B-2 | B-3 | B-4 | R-L | METEOR | CIDEr | Extreme |
|---|---|---|---|---|---|---|---|---|---|
| Blender -Vanilla | Base | 17.85 | 7.08 | 3.60 | 2.11 | 17.06 | 7.46 | 15.44 | **51.02** |
| | Aligned$_{major}$ | 19.07 | 7.71 | 3.94 | 2.28 | 17.09 | 7.71 | 15.97 | 50.61 |
| | Aligned$_{soft}$ | 17.88 | 7.21 | 3.68 | 2.12 | 16.52 | 7.31 | 15.50 | 50.73 |
| | Aligned$_{w/oA}$ | 19.70 | 7.56 | 3.64 | 2.05 | 16.90 | 7.72 | 15.62 | 50.48 |
| | Aligned$_{d-PM}$ | **20.75** | **8.32** | **4.17** | **2.39** | **17.41** | **8.21** | **16.57** | 50.38 |
| Blender -Joint | Base | 18.70 | 7.30 | 3.61 | 2.03 | 17.66 | 7.56 | 16.91 | 50.95 |
| | Aligned$_{major}$ | 20.37 | 8.61 | 4.47 | 2.65 | 19.23 | 8.32 | **21.86** | 51.57 |
| | Aligned$_{soft}$ | 19.36 | 7.87 | 3.85 | 2.09 | 17.55 | 7.65 | 15.90 | 50.84 |
| | Aligned$_{w/oA}$ | 21.05 | 8.14 | 3.89 | 2.07 | 17.65 | 8.11 | 15.29 | 50.68 |
| | Aligned$_{d-PM}$ | **21.05** | **8.97** | **4.74** | **2.78** | **19.39** | **8.48** | 20.34 | **51.81** |
| MultiESC | Base | 20.36 | 8.80 | 4.92 | 3.14 | 21.00 | 8.58 | 30.69 | 52.74 |
| | Aligned$_{major}$ | 19.10 | 8.27 | 4.61 | 2.88 | 20.72 | 8.24 | 30.15 | 52.57 |
| | Aligned$_{soft}$ | 19.30 | 8.33 | 4.62 | 2.88 | 20.83 | 8.35 | 30.75 | 52.54 |
| | Aligned$_{w/oA}$ | 21.58 | 8.80 | 4.74 | 2.96 | 20.47 | 8.78 | 28.58 | 51.65 |
| | Aligned$_{d-PM}$ | **21.59** | **9.56** | **5.33** | **3.36** | **21.50** | **9.03** | **32.65** | **53.15** |

**Experimental Setup**  We apply our proposed method to align each base model, thus treating the well-trained base model as the generator $\mathcal{G}(\xi_0)$. Additionally, to validate the effectiveness of d-PM, we employ three alternative preference models within our framework for comparative analysis:

(1) A preference model (major) trained to predict the majority voting result of annotations from different annotators, denoted as $l_m$, and optimized by cross-entropy loss, formulated as:

$$\mathcal{L}(\theta) = -\mathbb{E}_{(c,s,l_m)\sim\mathcal{D}}[p_l(l_m)\log(\mathcal{R}^{l_m}(c,s;\theta))]. \tag{10}$$

where $p_l(l_m)$ denotes the one-hot vector of $l_m$.

(2) A preference model (soft) trained to approximate the direct probabilistic label of annotations, i.e., the soft label $l$. The model is optimized by:

$$\mathcal{L}(\theta) = \mathbb{E}_{(c,s,l)\sim\mathcal{D}}\|\mathcal{R}(c,s;\theta) - l\|^2. \tag{11}$$

(3) A preference model (w/oA) that does not aggregate annotations and takes each annotation as independent. This model is optimized by cross-entropy loss, similar to Equation (10).

When training the aligned models, we aim to retain the same hyperparameters used in the training of the base models. We set the candidate number $K$ to 10. We train each aligned model five times with five different seeds. Subsequently, we test each of the five trained models on the test dataset and compute the average results.

**Automatic Evaluation**  We adopt the following metrics commonly used in previous work [5, 22] for the automatic evaluation of our proposed method: BLEU [27] (B-1/2/3/4), ROUGE (R-L) [20], METEOR [2], CIDEr [33], and BOW Embedding-based matching score [21] (Extreme). Results are shown in Table 1.

Our Aligned$_{d-PM}$ significantly improves the performance of the base model in almost all automatic metrics, irrespective of the base model, suggesting the overall effectiveness of our proposed method. Aligned$_{major}$ and Aligned$_{soft}$ are able to enhance the performance when the base model is either Blender-Vanilla or Blender-Joint, however, they do not yield an improvement when the base model is MultiESC. This limitation underscores the constraints inherent in using majority voting labels and soft labels to address disagreement of human preferences. Aligned$_{w/oA}$ surpasses the base model in certain metrics but falls short in others. Notably, it performs significantly lower in CIDEr, a metric evaluating the similarity between TFIDF-weighted n-grams. This shortfall suggests that Aligned$_{w/oA}$ is less likely to generate responses containing critical information found in the ground truth. This issue arises from the preference scores determined by w/oA being closely clustered in value, resulting in its inability to sequence the generated samples logically. These pieces of evidence indicate the potency of our proposed preference model d-PM.

Table 2: Human evaluation results on ESConv. The base model is MultiESC.

| Model | Identification | Comforting | Suggestion | Overall | Global Consensus |
|---|---|---|---|---|---|
| Base | 3.017 | 2.562 | 2.918 | 2.598 | 2.693 |
| Aligned$_{major}$ | 3.032 | 2.572 | 2.880 | 2.598 | 2.763 |
| Aligned$_{soft}$ | 3.007 | 2.557 | 2.905 | 2.568 | 2.747 |
| Aligned$_{d\text{-PM}}$ | **3.052** | **2.587** | **2.952** | **2.637** | **2.783** |

**Human Ratings**   We ask human annotators to evaluate the generations of models based on MultiESC since MultiESC can outperform Blender-Vanilla and Blender-Joint in almost all automatic evaluations. Specifically, we randomly sample 100 responses generated by different models for human ratings. We asked annotators to imagine they are help-seekers in the corresponding situation and measure each response in five aspects: (1). *Identification*: on a scale of $1 \sim 5$, how much the response can explore your situation in depth and help identify the problems. (2). *Comforting*: on a scale of $1 \sim 5$, how skillful the response can comfort you. (3). *Suggestion*: on a scale of $1 \sim 5$, how helpful the response can solve your problems. (4). *Overall*: on a scale of $1 \sim 5$, the overall quality of this response for emotional support; (5) *Global Consensus*: the number of people who deem the response can help them, $1 \sim 5$ represent nobody ($< 1\%$), rare ($5\% \sim 25\%$), controversial ($\sim 50\%$), most ($75\% \sim 90\%$), and all ($> 99\%$), respectively. Each response is annotated by three annotators, and we averaged these three annotations as the final result for each metric.

From Table 2, our method performs the best among the methods. Aligned$_{d\text{-PM}}$ obtained the highest score in all aspects, including the global consensus. It demonstrates that our method can generate less controversial and more helpful responses in the task of emotional support conversation.

## 4.2   Integrity RoT Generation

We also apply our method to the integrity "Rule-of-Thumb" (RoT) generation task. This task is concerned with describing a chatbot's normative rules, which holds great potential for advancing research on morally-consistent conversational agents [42]. When it comes to outlining a chatbot's normative rules, people's values can vary widely. However, morally-consistent conversational agents are expected to accommodate the values of as many individuals as possible. Therefore, the generation of widely acceptable RoTs is crucial for guiding the behavior of these agents.

**Dataset and Base Models**   The MIC dataset [42] comprises about 99k distinct RoTs that encapsulate the moral assumptions inherent in 38k machine-generated replies to open-ended prompts. Each prompt is associated with three different RoTs, each provided by a distinct annotator. Alongside each RoT, annotators offer a "global consensus" value, $\beta$, which signifies the estimated proportion of the global population that would agree with the RoT. We utilize T5 (small), Flan-T5 (base) and BART (large) models as our base models, and fine-tune them on the MIC dataset. For model inference, we closely follow the processes presented in [42]. Specifically, we adopt three decoding strategies: greedy decoding, beam search ($n = 3$), and nucleus sampling ($p = 0.9$). We generate one RoT for greedy decoding; for the latter two, three hypotheses are generated and the highest-scoring one is selected.

**Human Preferences with Disagreement**   We learn human preferences with disagreement for normative rules from the MIC dataset. The open-ended prompts are treated as context, and the ground truth RoT is considered the text to be evaluated. A probabilistic label is assigned to each RoT based on its global consensus: the probability for the class *acceptable* is $\beta$ while that for *unacceptable* is $(1 - \beta)$. We utilize this dataset to train the d-PM; details can be found in Appendix B.2.

**Experimental Setup**   We apply our proposed framework to align each base model that has been rigorously fine-tuned on the MIC dataset. To assess the effectiveness of d-PM, we also train a preference model (soft) by minimizing the loss computed using Equation (11). The number of candidates $K$ is set to 5. We adopt the same hyperparameters for training each aligned model as are used for the base model. We conduct five training runs for each aligned model using five distinct seeds. Then, we evaluate each of the five trained models on the test dataset and calculate the average results.

Table 3: Automatic evaluation results on MIC. † represents significantly better than the corresponding base model with $p < 0.01$.

| | Model | | R-1 | R-2 | R-L | BertScore | ScareBLEU | Avg.Len |
|---|---|---|---|---|---|---|---|---|
| T5 (Small) | Beam | Base | 53.44 | **32.97** | **52.17** | 93.44 | **29.05** | 8.94 |
| | | Aligned$_{soft}$ | 52.14 | 31.48 | 50.79 | 93.34 | 27.05 | 8.85 |
| | | Aligned$_{d\text{-}PM}$ | **53.45** | 32.82 | 52.09 | **93.49$^\dagger$** | 28.51 | **8.99** |
| | Greedy | Base | 37.04 | 16.30 | 35.27 | 90.94 | 14.27 | **10.47** |
| | | Aligned$_{soft}$ | 37.77$^\dagger$ | 16.82$^\dagger$ | 35.97$^\dagger$ | 91.21$^\dagger$ | 14.68$^\dagger$ | 9.83 |
| | | Aligned$_{d\text{-}PM}$ | **38.15$^\dagger$** | **17.22$^\dagger$** | **36.40$^\dagger$** | **91.29$^\dagger$** | **15.15$^\dagger$** | 9.74 |
| | $p$=0.9 | Base | 40.22 | 19.23 | 38.56 | 91.59 | 16.71 | **9.85** |
| | | Aligned$_{soft}$ | 40.90$^\dagger$ | 19.70$^\dagger$ | 39.24$^\dagger$ | 91.79$^\dagger$ | 16.98$^\dagger$ | 9.47 |
| | | Aligned$_{d\text{-}PM}$ | **41.41$^\dagger$** | **20.22$^\dagger$** | **39.75$^\dagger$** | **91.90$^\dagger$** | **17.75$^\dagger$** | 9.38 |
| Flan-T5 (Base) | Beam | Base | 55.07 | 34.96 | 53.74 | 93.77 | 30.68 | 9.00 |
| | | Aligned$_{soft}$ | 54.82 | 34.65 | 53.49 | 93.75 | 30.34 | 9.00 |
| | | Aligned$_{d\text{-}PM}$ | **55.18$^\dagger$** | **35.07$^\dagger$** | **53.86$^\dagger$** | **93.79$^\dagger$** | **30.83$^\dagger$** | **9.01** |
| | Greedy | Base | 37.94 | 17.23 | 36.13 | 91.39 | 15.36 | **9.78** |
| | | Aligned$_{soft}$ | 37.84 | 17.03 | 36.00 | 91.38 | 15.12 | 9.75 |
| | | Aligned$_{d\text{-}PM}$ | **38.34$^\dagger$** | **17.52** | **36.53$^\dagger$** | **91.44$^\dagger$** | **15.49** | 9.77 |
| | $p$=0.9 | Base | 41.41 | 20.41 | 39.70 | 92.02 | 18.02 | 9.30 |
| | | Aligned$_{soft}$ | 41.44 | 20.33 | 39.71 | 92.02 | 17.91 | 9.29 |
| | | Aligned$_{d\text{-}PM}$ | **41.78$^\dagger$** | **20.69$^\dagger$** | **40.09$^\dagger$** | **92.07$^\dagger$** | **18.26$^\dagger$** | **9.32** |
| BART (Large) | Beam | Base | 54.81 | 35.07 | 53.35 | 93.85 | 30.80 | **9.44** |
| | | Aligned$_{soft}$ | 54.82 | 34.85 | 53.36 | 93.82 | 30.35 | 9.36 |
| | | Aligned$_{d\text{-}PM}$ | **55.05$^\dagger$** | **35.18** | **53.62** | **93.86$^\dagger$** | **30.85** | 9.40 |
| | Greedy | Base | 54.77 | 34.85 | 53.30 | **93.84** | **30.51** | 9.54 |
| | | Aligned$_{soft}$ | 54.54 | 34.53 | 53.10 | 93.80 | 30.01 | 9.47 |
| | | Aligned$_{d\text{-}PM}$ | **54.81** | **34.86** | **53.39** | 93.83 | **30.51** | 9.48 |
| | $p$=0.9 | Base | 54.77 | 34.96 | 53.32 | 93.84 | 30.62 | **9.56** |
| | | Aligned$_{soft}$ | 54.65 | 34.68 | 53.23 | 93.82 | 30.16 | 9.45 |
| | | Aligned$_{d\text{-}PM}$ | **54.86** | **35.01** | **53.45** | **93.85$^\dagger$** | **30.63** | 9.48 |

Table 4: Human evaluation results on MIC. The base model is BART-large (beam).

| Model | Well-formedness | Fluency | Relevance | Global Consensus |
|---|---|---|---|---|
| Base | 0.528 | 2.547 | 2.037 | 2.428 |
| Aligned$_{soft}$ | 0.550 | **2.602** | 2.028 | 2.502 |
| Aligned$_{d\text{-}PM}$ | **0.568** | **2.602** | **2.103** | **2.555** |

**Automatic Evaluation**   In accordance with previous work [42], we report standard ROUGE [20] (R-1/2/L), ScareBLEU [27], BERTScore [38], and average generation length (Avg. Len) metrics. As each prompt-reply pair in our dataset has three ground truth RoTs, we compute each metric by taking the maximum score from these three, following the method employed by [42]. The results are displayed in Table 3.

The results clearly show that Aligned$_{d\text{-}PM}$ generally outperforms its base model. In addition, Aligned$_{d\text{-}PM}$ achieves the highest score across all evaluation metrics, except the generation length, when employing a beam decoding strategy. While Aligned$_{soft}$ slightly improves performance with T5 (small) as the base model, a minor decline is observed when the base model is either Flan-T5 (base) or BART (large). Interestingly, the enhancements observed in this task are not as pronounced as those witnessed in the emotional support conversation task (refer to Table 1). This may be attributed to the MIC dataset inherently accounting for disagreement, as each prompt is paired with three RoTs from different annotators. This enables base models, when fine-tuned on this dataset, to encapsulate various human preferences to a certain degree. Nonetheless, our framework has the potential to further boost model performance by providing more explicit preference information with disagreement during training.

**Human Ratings**   We randomly select 100 replies generated by models with BART (large) as the base model for human evaluation. Adhering to previous practice, we assess generated outputs based on the following criteria [42]: (1). *Well-formedness*: yes or no, does the RoT explain the basics of good or bad behavior with a single judgment or action?; (2). *Fluency*: on a scale of 1-5, how much does the RoT align with what an English speaker might naturally say?; and (3) *Relevance*: on a scale of 1-5, how well does the RoT apply to the Answer for this specific Question if we assume the RoT is true? Furthermore, we request annotators to provide a *Global Consensus*: how many people globally will agree with this RoT, similar to the method described in Section 4.1. Three annotators evaluate each RoT, and we average these three evaluations as the final score for each metric.

Results presented in Table 4 indicate that our method produces RoTs that are more universally agreeable than those generated by the other two models. Our Alignd-PM model improves all metrics over the base model and outperforms Alignedsoft.

## 5   Model Analysis

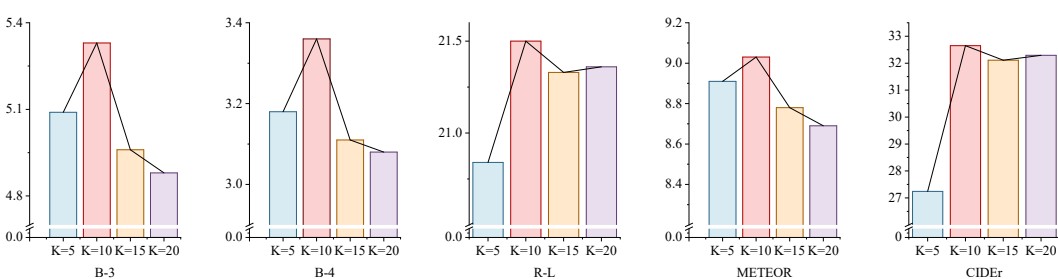

Figure 4: Model performances with different candidate numbers $K$ when calibrating MultiESC with preference scores of d-PM.

**The effect of candidate number $K$ during calibration.**   To examine the influence of varying the candidate number $K$ on model calibration, we modify the MultiESC calibration process using different candidate numbers, namely 5, 10, 15, and 20. Specifically, this involves changing the beam widths in the diverse beam search process. Theoretically, a more significant candidate number would encompass more samples under consideration, thereby escalating the upper bound of performance. Nevertheless, as depicted in Figure 4, the model performance initially experiences an augmentation yet subsequently exhibits a reduction with the increment of the variable $K$. It is because an overly large candidate number could introduce redundant samples with minor differences, and the generation model might erroneously distinguish between them.

| Metric | MultiESC | RL | Ours |
|--------|----------|-------|-------|
| **B-1** | 20.36 | 11.75 | 21.59 |
| **B-2** | 8.80 | 4.81 | 9.56 |
| **B-3** | 4.92 | 2.78 | 5.33 |
| **B-4** | 3.14 | 1.81 | 3.36 |
| **R-L** | 21.00 | 19.57 | 21.50 |
| **METEOR** | 8.58 | 6.25 | 9.03 |
| **CIDEr** | 30.69 | 26.02 | 32.65 |
| **Extreme** | 52.74 | 51.21 | 53.15 |
| **#(Samples)/s** | - | 2.65 | 4.36 |

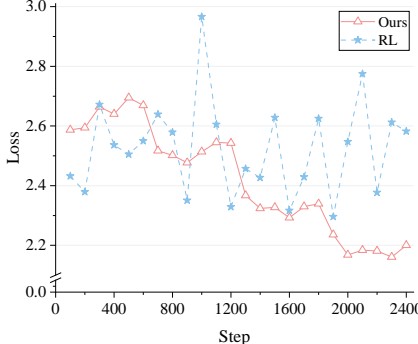

Figure 5: Comparison between alignment with RL (RL) and our model (Ours). Left: Automatic evaluation results (#(Samples)/s indicates the number of trained samples per second). Right: Training loss according to training steps.

**Contrastive Learning vs. Reinforcement Learning.** We opt for contrastive learning to align generation models with human preferences instead of the currently prevalent reinforcement learning (RL) approach. This decision is rooted in several considerations. Firstly, RL requires expensive online decoding procedures, while our framework based on contrastive learning is a one-time offline process [40]. Secondly, RL usually yields a slow convergence speed [9]. To validate this, we align MultiESC using RL. From Figure 5, RL trains fewer samples than our framework per second. After the same steps, the loss of RL is still high, and the model performance is much worse than ours.

## 6  Related Work

Developing human-centric systems, which ensure that human stakeholders benefit from system outcomes [14], remains a great challenge. To build a human-centric system, it is critical to align models with human preferences [35]. There are various methods to implement the alignment. Currently, the most popular and well-known method is reinforcement learning from human feedback, thanks to the GPT series [26]. This method is also used for text summarization [31, 4], detoxification [6], and machine translation [15, 16]. The above-mentioned methods adopt one reward model, while some methods combine rewards computed by different reward models to consider fine-grained aspects of human needs [8, 11]. Moreover, some models use human feedback as the supervision signal directly to learn human preferences, such as fine-tuning pre-trained models with well-established datasets [10]. Human feedback can also be used to augment prompts for better performance [24].

## 7  Conclusion and Future Work

In this work, we strive to align models with human preferences to foster the development of trustworthy human-centric NLG systems. Unlike previous approaches, we take inherent disagreement into account for modeling human preferences. This idea is motivated by two compelling reasons. Firstly, it is impractical to expect consensus in human preferences due to the high degree of subjectivity involved. Secondly, harmonizing preference disagreements can inadvertently disadvantage minority groups. Accordingly, we introduce a Bayesian approach, termed Preference Modeling with Disagreement (short as **d-PM**), to capture the subtleties of disagreement from limited human feedback. We subsequently utilize its preference scores to calibrate pre-existing text generation models. The efficacy of our method is substantiated through experiments in emotional support conversation and integrity Rule-of-Thumb generation.

Despite our focus on disagreement, another critical aspect remains to consider when modeling human preferences. In this work, like most previous studies, we assumed a linear relationship between the preference score and the proportion of the global population that finds the text acceptable. However, this assumption may not always hold true. For example, a sentence that $20\%$ of people find helpful should ideally have a preference score closer to $0$ instead of $0.2$. We believe that this issue merits further exploration in future research. Fortunately, in this study, we sidestepped this problem by using the rank of preference scores rather than the scores themselves.

## 8  Acknowledgements

This work is supported by Research Grants Council of Hong Kong (PolyU/5204018, PolyU/15207920, PolyU/15213323) and National Natural Science Foundation of China (62302184, 62076212).

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

# A  Ethics and Societal Impact

**Ethics**  In our experiments, we utilized three datasets: MI-dataset, ESConv, and MIC. Each of these datasets is designed to protect user privacy, and none contain any personal information.

**Societal Impact**  Our work is driven by the goal to enhance the positive impact of Natural Language Generation (NLG) models by better aligning them with human preferences, particularly those relating to personal standards and human values such as emotions, beliefs, and ethical norms. We acknowledge the diversity of individual preferences, which can often lead to disagreements. This disagreement is taken into account in our model of human preferences, which we use to calibrate our text generation model. This approach aims to produce text that is widely acceptable and less controversial.

Our findings suggest that our approach can make generated texts less contentious and more useful. Looking towards the future, neglecting to account for disagreements in human preferences could result in serious implications, especially if these models are used in safety-critical situations. We anticipate that our approach could be beneficially applied to a wider range of tasks and models in future research and product development.

# B  Supplement Experiment Details

The code for our experiments is provided in the supplements and will be made public upon acceptance of this paper. Additional details that are not included in the submitted code are explained in the following subsections.

## B.1  Resources Used

We trained models based on MultiESC using two Nvidia RTX 3092 GPUs, while all other models, including d-PM models, were trained using a single NVIDIA RTX 3092 GPU. Our models were implemented in Python using PyTorch[3] and the transformers (4.16.2) library[4].

## B.2  d-PM Training Details

**Model Architecture**  The d-PM models are fine-tuned versions of the BERT (base, uncased) model using prefix-tuning. The BERT model has a layer number of $n_{\text{layer}}^{\text{BERT}}$, and a hidden size of $d^{\text{BERT}}$. The model uses a sequence of prefix indices, represented as $\text{L}_{\text{idx}}$, and the length of the prefix is represented by $|\text{L}_{\text{idx}}|$. The model input is a concatenation of the prefix, the context, and the text, denoted as $z = [\text{L}_{\text{idx}}; c; s]$, where $z_t$ represents the input at time step $t$. The activations of the model are denoted as $h = [h^{\text{prefix}}, h^c, h^s]$, where $h_t$ represents the concatenation of all activation layers at time step $t$. Here, $h^c$ and $h^s$ are computed using BERT parameters, while $h^{\text{prefix}}$ are free parameters. A trainable matrix $A_{\theta_1}$ is then initialized, which is parameterized as $\theta_1$ of dimension $|\text{L}_{\text{idx}}| \times d^{\text{dim}}$, where $d^{\text{dim}} = n_{\text{layer}}^{\text{BERT}} \times d^{\text{BERT}}$. Thus, the activation $h_t$ at time step $t$ is defined as:

$$h_t = \begin{cases} A_{\theta_1}[t,:] & \text{if } t < |\text{L}_{\text{idx}}|, \\ \text{BERT}(z_t, h_{<t}) & \text{otherwise.} \end{cases} \tag{12}$$

We utilized the last hidden states (the activations from the last layer) of the text $s$ to calculate the probabilities for the classes *acceptable* and *unacceptable*. This computation is facilitated by a multilayer perceptron (MLP) layer, which is parameterized as $\theta_2$.

The parameters of the d-PM, i.e., $\theta = (\theta_1, \theta_2)$, are optimized using the loss function, which is formulated in Equation (6). Furthermore, the preference models that we utilized in Alignedmajor and Alignedsoft share the same architecture. However, they are differentiated by the distinct loss functions used for their optimization, as denoted in Equation (10) and Equation (11), respectively.

**Experimental Setup**  The d-PM models were implemented based on the code[5] published for the UnifiedSKG framework [37]. During the training of the d-PM models, BERT parameters were frozen,

---

[3] https://pytorch.org/
[4] https://huggingface.co/docs/transformers/v4.16.2/en/index
[5] https://github.com/HKUNLP/UnifiedSKG

and the prefix was optimized. The prefix length was set to 10. A batch size of 160 and a learning rate of $5 \times 10^{-4}$ were used.

**Dataset**    For the training dataset, the MI-Dataset was used to train d-PM model which can output human preferences for the emotional support responses. This dataset contains 15 fine-grained classes of responses. The annotators categorized each response into one of the fine-grained classes. These responses fall into three coarse-grained classes according to the MI codes [25]: (1) *MI Adherent*, which involves supporting the help-seekers with empathetic and compassionate statements, enabling them to feel heard, respected, and understood; (2) *Relational*, which focuses on establishing a solid relationship between the help-seeker and the supporter, leading to more positive responses from the help-seekers; and (3) *MI Non-Adherent*, which includes statements such as arguing, confronting, or offering unsolicited advice, and may create resistance and hinder problem-solving for the help-seekers. When training the d-PM, these classes were converted into two classes: *acceptable* and *unacceptable*, as shown in Table 5 The dataset was randomly split into a $9 : 1$ ratio for the training and validation set. There are $34.93\%$ instances where annotators have different annotations.

Table 5: The MI codes in MI-Dataset [36], and our recategorization. Examples are from its paper.

| Our Classes | Coarse-Grained MI Codes | Fine-Grained MI Codes | Examples |
|---|---|---|---|
| *acceptable* | MI Adherent | 1. Advise with Permission | If you agree with it, we could try to brainstorm some ideas that may help. |
| | | 2. Affirm | You should be proud of yourself for your past efforts. |
| | | 3. Emphasize Autonomy | It is really up to you to decide. |
| | | 4. Support | I know it's really hard to stop drinking. |
| | Relational | 5. Closed Question | Do you think this is an advantage? |
| | | 6. Open Question | What is your take on that? |
| | | 7. Simple Reflection | It sounds like you're feeling worried. |
| | | 8. Complex Reflection | **Speaker:** Mostly, I would change for future generations. **Listener:** It sounds like you have a strong feeling of responsibility |
| | | 9. Give Information | Logging your cravings is important as cravings often lead to relapses. |
| | | 10. Self-Disclose | I used to be similar where I get obsessed about how people look. |
| | | 11. Other | Good morning, Hi there. |
| *unacceptable* | MI Non-Adherent | 12. Advise without Permission | You should simply scribble a note that reminds you to take a break. |
| | | 13. Confront | Yes, you are an alcoholic. You might not think so, but you are. |
| | | 14. Direct | Don't do that! |
| | | 15. Warn | Be careful, DO NOT stop taking meds without discussing with your doctor. |

Table 6: One example from the MIC dataset. **Blue text**: prompt-reply (QA) pair. **Orange text**: global consensus scale value.

| |
|---|
| **Question**: Am I a bad BF, weird, or just going insane? |
| **Answer**: I don't think you're a bad bf or weird or going insane. I think you just need to talk to him about it. |
| **RoT**: It's important to communicate honestly with your significant other. |
| **Global Consensus**: 4 |

The MIC dataset was used for training the d-PM used in the integrity Rule of Thumb (RoT) generation task. Each RoT in this dataset is annotated with a global consensus provided in a 5-Likert scale format, where $1 \sim 5$ represent nobody ($< 1\%$), rare ($5\% \sim 25\%$), controversial ($\sim 50\%$), most

$(75\% \sim 90\%)$, and all $(> 99\%)$, respectively. An example is shown in Table 6. For the purpose of training, the ordinal scales were converted into continuous variables that represent the percentage consensus. This was done by randomly selecting a value between the upper and lower bound corresponding to each scale value to serve as the final global consensus ($\beta$) for each RoT. Following this, the probability for the class *acceptable* was set as $\beta$, while that for *unacceptable* was set as $(1 - \beta)$. The split of the MIC dataset was maintained when training the model and when calibrating the generation model to avoid data leakage.

### B.3 Emotional Support Conversation

**Experimental Setup**   We implemented the base models following the methods detailed in [22, 5] and their corresponding published codes[6]. Each base model was run once using the seed provided in its published codes.

In order to fairly compare models, we aimed to keep the hyperparameters consistent with those of the corresponding base models when implementing aligned models, except that we increased the learning rate to enhance the training efficiency of aligned models. Specifically, we set the learning rate to $1 \times 10^{-3}$ for the Blender-Vanilla and Blender-Joint base models, and $3 \times 10^{-5}$ for the other models. Additionally, due to GPU memory constraints, we reduced the batch size from 32 to 12 when training the aligned MultiESC. We ran each aligned model five times with different seeds: 0, 1, 13, 42, and 1024, and reported the average results.

### B.4 Integrity RoT Generation

**Experimental Setup**   Our implementation of the base models was based on the work presented in the paper [42] and its published codes [7]. Each base model was run once using the seed provided in its published codes.

For a fair comparison, we endeavored to keep the hyperparameters consistent with those of the corresponding base models when implementing the aligned models. However, we increased the learning rate to $1 \times 10^{-4}$ during the training of aligned models. Each aligned model was run five times with different seeds: 0, 1, 13, 42, and 1024, and we reported the average results.

### B.5 Alignment through Reinforcement Learning

**Implementation Details**   The reinforcement learning (RL) framework was implemented with reference to the Hugging Face TRL library[8]. We aimed to ensure consistency with the hyperparameters used in the aligned model, which was implemented using the contrastive learning framework. Additionally, we adhered to the same training steps, i.e., terminating the training process after 2400 steps. We executed the RL framework with five seeds: 0, 1, 13, 42, and 1024. The best result obtained is reported in the paper.

**Full Results**   Table 7 and Table 8 present results of aligning models with RL framework. Generally, alignment with RL falls short in performance when compared to the contrastive learning (CL) framework, i.e., our proposed model. In the task of emotional support conversation, as depicted in Table 7, aligning with RL proves to be incompatible with ours and can occasionally result in a degradation in the base model's performance. As for the task of RoT generation, as shown in Table 8, the best performance when aligning each base model is nearly always achieved by our model. Alignment with RL notably increases evaluation metric values of T5 (small) and Flan-T5 (base) when employing a greedy or $p = 0.9$ decoding strategy. However, a common issue among models aligned with RL is their tendency to generate phrases like "(something) is good/wrong" and "It is good/wrong to (something)". Although the key information (something) may be incorrect, such a pattern can exhibit significant overlap in wording with the ground truth, thereby yielding higher values in terms of some evaluation metrics. The potential reason for the suboptimal performance of RL could be the small size of ESConv and MIC. Although RL has proven potent in certain scenarios, as illustrated by

---

[6]Blender-Vanilla/Joint: `https://github.com/thu-coai/Emotional-Support-Conversation`; MultiESC: `https://github.com/lwgkzl/MultiESC`.

[7]`https://github.com/SALT-NLP/mic`

[8]`https://huggingface.co/docs/trl/index`

Table 7: Automatic evaluation results on ESConv. † represents significantly better than the corresponding base model with $p < 0.01$.

| Model | | B-1 | B-2 | B-3 | B-4 | R-L | METEOR | CIDEr | Extreme |
|---|---|---|---|---|---|---|---|---|---|
| Blender -Vanilla | Base | 17.85 | 7.08 | 3.60 | 2.11 | 17.06 | 7.46 | 15.44 | **51.02** |
| | RL | 19.31 | 7.01 | 3.19 | 1.70 | 15.64 | 7.54 | 12.85 | 50.10 |
| | Ours | **20.75**† | **8.32**† | **4.17**† | **2.39**† | **17.41**† | **8.21**† | **16.57**† | 50.38 |
| Blender -Joint | Base | 18.70 | 7.30 | 3.61 | 2.03 | 17.66 | 7.56 | 16.91 | 50.95 |
| | RL | **21.26** | 8.43 | 4.12 | 2.29 | 17.69 | 8.32 | 16.96 | 51.48 |
| | Ours | 21.05† | **8.97**† | **4.74**† | **2.78**† | **19.39**† | **8.48**† | **20.34**† | **51.81**† |
| MultiESC | Base | 20.36 | 8.80 | 4.92 | 3.14 | 21.00 | 8.58 | 30.69 | 52.74 |
| | RL | 21.58 | 8.80 | 4.74 | 2.96 | 20.47 | 8.78 | 28.58 | 51.65 |
| | Ours | **21.59**† | **9.56**† | **5.33**† | **3.36**† | **21.50**† | **9.03**† | **32.65**† | **53.15**† |

Table 8: Automatic evaluation results on MIC. † represents significantly better than the corresponding base model with $p < 0.01$.

| Model | | | R-1 | R-2 | R-L | BertScore | ScareBLEU | Avg.Len |
|---|---|---|---|---|---|---|---|---|
| T5 (Small) | Beam | Base | 53.44 | **32.97** | **52.17** | 93.44 | 29.05 | 8.94 |
| | | RL | 52.64 | 32.44 | 51.68 | 93.21 | **29.19** | 7.42 |
| | | Ours | **53.45** | 32.82 | 52.09 | **93.49**† | 28.51 | **8.99** |
| | Greedy | Base | 37.04 | 16.30 | 35.27 | 90.94 | 14.27 | **10.47** |
| | | RL | **40.04**† | **19.71**† | **38.86**† | **91.67**† | **18.67**† | 7.45 |
| | | Ours | 38.15† | 17.22† | 36.40† | 91.29† | 15.15† | 9.74 |
| | $p$=0.9 | Base | 40.22 | 19.23 | 38.56 | 91.59 | 16.71 | **9.85** |
| | | RL | **42.84**† | **22.28**† | **41.69**† | **92.10**† | **20.74**† | 7.41 |
| | | Ours | 41.41† | 20.22† | 39.75† | 91.90† | 17.75† | 9.38 |
| Flan-T5 (Base) | Beam | Base | 55.07 | 34.96 | 53.74 | 93.77 | 30.68 | 9.00 |
| | | RL | 50.84 | 30.95 | 49.78 | 93.18 | 27.95 | 7.30 |
| | | Ours | **55.18**† | **35.07**† | **53.86**† | **93.79**† | **30.83** | **9.01** |
| | Greedy | Base | 37.94 | 17.23 | 36.13 | 91.39 | 15.36 | **9.78** |
| | | RL | **42.97**† | **22.36**† | **41.54**† | **92.27**† | **20.71**† | 7.30 |
| | | Ours | 38.34† | 17.52 | 36.53† | 91.44† | 15.49 | 9.77 |
| | $p$=0.9 | Base | 41.41 | 20.41 | 39.70 | 92.02 | 18.02 | 9.30 |
| | | RL | **45.61**† | **25.11**† | **44.26**† | **92.64**† | **22.97**† | 7.27 |
| | | Ours | 41.78† | 20.69† | 40.09† | 92.07† | 18.26† | **9.32** |
| BART (Large) | Beam | Base | 54.81 | 35.07 | 53.35 | 93.85 | 30.80 | **9.44** |
| | | RL | 53.02 | 33.22 | 51.58 | **94.26** | 29.91 | 8.39 |
| | | Ours | **55.05**† | **35.18** | **53.62** | 93.86† | **30.85** | 9.40 |
| | Greedy | Base | 54.77 | 34.85 | 53.30 | 93.84 | **30.51** | **9.54** |
| | | RL | 53.30 | 33.30 | 51.83 | **94.28** | 29.88 | 8.52 |
| | | Ours | **54.81** | **34.86** | **53.39** | 93.83 | **30.51** | 9.48 |
| | $p$=0.9 | Base | 54.77 | 34.96 | 53.32 | 93.84 | 30.62 | **9.56** |
| | | RL | 53.36 | 33.37 | 51.89 | **94.29** | 29.96 | 8.55 |
| | | Ours | **54.86** | **35.01** | **53.45** | 93.85† | **30.63** | 9.48 |

recent research [41, 31, 16], a commonality among these successful applications is their training on expansive datasets (280K to 6M training instances). In contrast, the sizes of ESConv and MIC are significantly more modest. This hypothesis also aligns with findings from Agarwal et al. [1], which suggest that smaller datasets can potentially undermine RL's performance.

## C Case Study

We present case studies for the task of emotional support conversation and integrity RoT generation in Table 9 and Table 10, respectively.

Table 9: **Green text**: the help-seeker's emotion and problem. **Blue text**: the conversation context (history). **Orange text**: human evaluation results.

---

********** *Case One* **********

**The Help-seeker's Emotion**: Anxiety

**The Help-seeker's Problem**: Need a second job to pay off credit card bills.

**Problem Type**: Job crisis

| | | | |
|---|---|---|---|
| **Supporter**: Hello, how are you today ? | | | |
| **Seeker**: I'm good, busy. I need another job to pay more bills . | | | |
| **Base** | I'm sorry to hear that. Is there anything you'd like to talk about? | | |
| Overall | 2,1,2 | Global Consensus | 2,2,2 |
| **Aligned$_{major}$** | I'm sorry to hear that. Would you like to talk about it? | | |
| Overall | 3,2,2 | Global Consensus | 3,3,2 |
| **Aligned$_{soft}$** | I'm sorry to hear that. Would you like to talk about it? | | |
| Overall | 3,2,2 | Global Consensus | 3,3,2 |
| **Aligned$_{d-PM}$** | I'm sorry to hear that. Is there a way you can get a second job to pay off your debts? | | |
| Overall | 4,5,4 | Global Consensus | 4,4,5 |

********** *Case Two* **********

**The Help-seeker's Emotion**: Anxiety

**The Help-seeker's Problem**: Applying for work, I have been retired for 2 years and would like to return to the workforce.

**Problem Type**: Procrastination

| | | | |
|---|---|---|---|
| **Seeker**: I have been retired for over 2 years and would like to return to the workforce, part-time. | | | |
| **Supporter**: So you are thinking of going back to work? | | | |
| **Seeker**: Yes, it would help us greatly. | | | |
| **Supporter**: You have the right idea already. | | | |
| **Seeker**: There are plenty of jobs and not sure where to start. | | | |
| **Base** | That is a great idea. I think it would be great to get your mind off of retirement. | | |
| Overall | 2,3,3 | Global Consensus | 2,4,3 |
| **Aligned$_{major}$** | Yes, there are plenty of jobs out there. You just need to know where to start. | | |
| Overall | 4,4,3 | Global Consensus | 3,4,3 |
| **Aligned$_{soft}$** | I see. So you are thinking about getting back into the field? | | |
| Overall | 4,4,4 | Global Consensus | 4,4,4 |
| **Aligned$_{d-PM}$** | That is a very good question. Would you be open to looking for a new job? | | |
| Overall | 4,5,4 | Global Consensus | 4,4,4 |

********** *Case Three* **********

**The Help-seeker's Emotion**: Sadness

**The Help-seeker's Problem**: I have a friend who knew me since I was little, she is closer to my mother's age so she is kinda like my aunt growing up. She always treat me like I was little. But I'm a grown woman now and I want our relationship to go far and grow more.

**Problem Type**: Problems with friends

| | | | |
|---|---|---|---|
| **Seeker**: I have a friend who knew me since I was little. She is a little older than me. She still treats me like I'm still little. I tried to explain to her that I'm a grown woman and I grew out of a lot of things when I was a kid. | | | |
| **Supporter**: Is she listening to you? I know how it feels to be patronized by people who are older. | | | |
| **Seeker**: She still didn't get it. I want our relationship to move further but with her like this. I feel sad. Is there anything I can do? | | | |
| **Supporter**: Well, if she is not listening to you when you try to talk to her you have to ask yourself whether the relationship is really worth it? | | | |
| **Seeker**: That's true . I didn't think about it. I know she is a good person. So I would give her a chance and we can talk about it before. | | | |
| **Base** | I think it's always worth talking to someone who is older than you. | | |
| Overall | 2,1,3 | Global Consensus | 2,3,3 |
| **Aligned$_{major}$** | Well, it's always worth talking to someone who has been through a similar situation. | | |
| Overall | 3,4,2 | Global Consensus | 3,3,2 |

| **Aligned$_{soft}$** | Well, it's always worth talking to someone who has been through a similar situation. | | |
|---|---|---|---|
| Overall | 3,4,2 | Global Consensus | 3,3,2 |
| **Aligned$_{d\text{-PM}}$** | I think that's a great idea. I think it's great that you are willing to give it a try. | | |
| Overall | 4,5,5 | Global Consensus | 4,5,4 |

Table 10: RoTs generated by different models, whose base model is BART-Large. **Blue text**: prompt-reply (QA) pair. **Orange text**: human evaluation results.

| ********** *Case One* ********** | | | |
|---|---|---|---|
| **Question**: Is it weird that I don't want a family or a career? | | | |
| **Answer**: No, it's not weird at all. It's perfectly normal to not want children or a family. | | | |
| **Base** | It is okay to not want children or a family. | Well-formedness | 1,1,1 |
| | | Relevance | 4,3,5 |
| | | Global Consensus | 3,3,4 |
| **Aligned$_{soft}$** | It is good to have a family. | Well-formedness | 0,0,1 |
| | | Relevance | 3,2,5 |
| | | Global Consensus | 1,2,3 |
| **Aligned$_{d\text{-PM}}$** | It is okay not to want a family or a career. | Well-formedness | 1,1,1 |
| | | Relevance | 4,4,5 |
| | | Global Consensus | 5,4,4 |
| ********** *Case Two* ********** | | | |
| **Question**: Who was the best teacher you ever had? What qualities made them great teachers? | | | |
| **Answer**: I don't know. I guess he had a lot of patience and he made me feel safe. | | | |
| **Base** | It is good to have a good teacher. | Well-formedness | 0,1,0 |
| | | Relevance | 2,2,1 |
| | | Global Consensus | 4,3,2 |
| **Aligned$_{soft}$** | It is good to have a good teacher. | Well-formedness | 0,1,0 |
| | | Relevance | 2,2,1 |
| | | Global Consensus | 4,3,2 |
| **Aligned$_{d\text{-PM}}$** | It is good for teachers to be patient with their students. | Well-formedness | 1,1,1 |
| | | Relevance | 5,4,4 |
| | | Global Consensus | 4,4,3 |
| ********** *Case Three* ********** | | | |
| **Question**: HELP, a hotel sink fell into the cupboard. Am I responsible or liable? Should I try to fix it or report it to the desk? HELP | | | |
| **Answer**: I would call the front desk and let them know. They should be able to help you. | | | |
| **Base** | It is good to ask for help when you need it. | Well-formedness | 1,1,0 |
| | | Relevance | 1,2,2 |
| | | Global Consensus | 3,4,2 |
| **Aligned$_{soft}$** | It is good to report maintenance issues to the proper authorities. | Well-formedness | 1,1,0 |
| | | Relevance | 3,3,1 |
| | | Global Consensus | 3,3,1 |
| **Aligned$_{d\text{-PM}}$** | It is good to report broken things to the hotel desk. | Well-formedness | 1,1,1 |
| | | Relevance | 3,3,4 |
| | | Global Consensus | 4,4,5 |

## D  Human Rating Details

**Annotation Details**  We engaged six annotators from diverse educational, gender, geographic, and occupational backgrounds to conduct human ratings for each task. Each sentence to be rated was assigned to three annotators, and the final result was calculated as the average of their scores. Given

that $3 \sim 4$ sentences were generated by different models for the same input, we presented these sentences and their corresponding context or prompt-reply pair on a single page. Annotators were asked to score the generated sentences relative to each other. To avoid potential bias, the order of sentences was randomized to prevent annotators from identifying the generation model based on sentence order.

**Annotation Instructions**    Annotators were initially briefed on the tasks to ensure they understood the objectives of the emotional support responses or RoTs. Except for the global consensus, they were asked to rate the generated responses according to their personal preferences. For the emotional support conversation task, we asked annotators to imagine themselves as help-seekers in the conversations. After providing the situational and emotional context, annotators then rated the generated responses.

**Annotation Cost**    We remunerated annotators at a rate of $2 \sim 3$ times their local hourly minimum wage. According to the annotators' feedback, they spent approximately $40$ seconds evaluating each generated response or RoT.

