# OpenReview forum: "Aligning Language Models with Human Preferences via a Bayesian Approach"
_NeurIPS.cc/2023/Conference — NeurIPS 2023 poster_

### Official Review · Reviewer_mWz9 · 2023-07-05

**Soundness:** 3 good
**Presentation:** 2 fair
**Contribution:** 2 fair
**Rating:** 4
**Confidence:** 4

**Summary:**

This paper proposes a method for modeling human preference with a Bayesian approach that considers the disagreement among human annotators. The authors also propose a contrastive learning method to replace RL in the alignment process. The authors conduct experiments on two text generation tasks and show some improvements upon baselines.

**Strengths:**

1. The paper is well-motivated and making use of disagreements among human preferences is an interesting problem.
2. The proposed method is technically sound. Using Bayesian and variational inference for modeling uncertainty is a good idea and makes sense to me.
3. The authors conduct experiments on two datasets and some improvements are observed compared to the baselines.
4. The authors conduct human evaluation on the evaluation datasets and also some ablation studies / analysis, which are helpful to understand the method better.

**Weaknesses:**

1. The experiments are not solid enough. First, the backbone models used in both the datasets are not strong enough. And it is hard to use human feedback when the model capacity is very limited. The performance improvements are also very marginal, which may come from some regularization effects in the contrastive learning objective. The poor performance from RL baseline also validate this hypothesis. It is also very strange why some baselines with major/soft voting lead to worse performance compared to the baseline. Maybe due to some bad hyperparameters? How are the hyperparameters tuned?
Also, the second dataset's human preference data is not actual collected from human annotation, which does not makes sense. It is unclear why the authors do not use some recent RLHF datasets.

2. The claim about RL's disadvantages is not necessarily true. Using policy gradient or PPO is computationally more efficient because it does not require decoding multiple outputs for contrastive learning, and both the proposed method and RL-based method require using the reward function to score the output. And the success of RLHF algorithms clearly shows the usefulness of RL-based method. It would be helpful to combine RL + the reward function trained with the Bayesian method.


**Questions:**

It would be interesting to see the actual difference of the preference model trained by conventional approaches and the proposed Bayesian approach. Can it be validated on some held out data that have disagreement annotations?

**Limitations:**

Yes

---

> ### Author Rebuttal · Authors · 2023-08-10
>
> $\textbf{Mistakes in W1, Q1:}$ The second dataset's human preference data is not actual collected from human annotation, which does not makes sense. \& Can it be validated on some held out data that have disagreement annotations?
>
> To clarify, the human preference data from the MIC dataset is genuinely derived from actual human annotations. The MIC dataset consists of a prompt-reply pair, with the prompt sourced from Reddit and the reply generated by a model. Furthermore, it includes a $\textbf{human-written}$ RoT and a global consensus, both ascertained by the same human annotator.
>
> It is imperative to highlight that while the dataset contains model-generated content, this is not our primary generation target and is distinct from human preference. We will make sure to communicate this more explicitly in our revised manuscript.
>
> Both datasets (MI dataset and MIC) contain human disagreement.
>
> $\textbf{W1.1:}$ It is also very strange why some baselines with major/soft voting lead to worse performance compared to the baseline. Maybe due to some bad hyperparameters? How are the hyperparameters tuned?
>
> Given the inherent subjectivity in our data, majority or soft voting methods may not necessarily capture the nuanced disagreements present. This could lead to an overly biased model towards a specific preference, missing out on the richness of diverse perspectives.
>
> As stated in the paper (lines 493, 505), we predominantly stick to the hyperparameters of the base models, with modifications made only to the learning rate and batch size.
>
> $\textbf{W1.2:}$ It is unclear why the authors do not use some recent RLHF datasets.
>
> We appreciate your feedback. Our selection of datasets was primarily driven by two key criteria: 1) the presence of high subjectivity, and 2) the potential to capture human disagreement in evaluations. We aimed to stress-test our approach in scenarios where diverse human perspectives might lead to varying judgments. While there are recent RLHF datasets available, many of them are centered around domains where human subjectivity doesn't introduce significant disagreement. Therefore, our chosen datasets better serve the purpose of validating our core idea.
>
> $\textbf{W2:}$ The claim about RL's disadvantages is not necessarily true. Using policy gradient or PPO is computationally more efficient because it does not require decoding multiple outputs for contrastive learning, and both the proposed method and RL-based method require using the reward function to score the output. And the success of RLHF algorithms clearly shows the usefulness of RL-based method. It would be helpful to combine RL + the reward function trained with the Bayesian method.
>
> Thank you for bringing this to our attention. While it's true that PPO doesn't necessitate decoding multiple outputs within a single epoch, the overall number of decodings is contingent upon the total number of training epochs. In our own implementation of RL, we observed that the convergence rate tends to be slower, necessitating a higher number of epochs and, consequently, more decoding instances. This, in turn, could offset the computational advantages gained in a single epoch. In our implementation, we obtained the best results after $25$ epochs. We will state this clearly in our paper.

---

> > ### Comment · Area_Chair_j9oz · 2023-08-18
> > **Please Reply to Author Rebuttal**
> >
> > Dear reviewer,
> >
> > Thanks a lot for your efforts and valuable reviews. Would you please check this author rebuttal and see how they address your concerns on experiment and claim on RL disadvantages? Please reply to authors by adding your following comments below this author rebuttal.
> >
> > As the author-reviewer discussion is closed soon, we would appreciate if you could submit your reply to authors by Aug 21st 1pm EDT.
> >
> > Thanks!
> >
> > Best,
> > AC

---

> ### Author Response · Authors · 2023-08-21
> **Additional RL Experiments in "Author Rebuttal by Authors"**
>
> We would like to kindly draw your attention to the ``Author Rebuttal by Authors`` section, specifically the third point. In that section, we have added the results of our RL-based alignment (**Table 3**), which indicates the superiority of our framework to RL. We believe these results address some of the concerns raised and provide additional context to our findings.
>
> While Reinforcement Learning (RL) has proven potent in certain scenarios, as illustrated by recent research [1,2,3], a commonality among these successful applications is their training on expansive datasets (280K to 6M training instances). In contrast, our dataset's size is significantly more modest. It is possible that this small data size plays a role in the suboptimal performance of RL in our experiments. This hypothesis aligns with findings from Agarwal et al. [4], which suggest that smaller datasets can potentially undermine RL's performance.
>
> We kindly request that you review the updated RL experiment results in the rebuttal. We hope this additional information can alleviate your concerns about **W2** and will be beneficial for evaluating our paper.
>
> Thank you for your time and consideration.
>
> [1] Ziegler D M, Stiennon N, Wu J, et al. Fine-tuning language models from human preferences[J]. arXiv preprint arXiv:1909.08593, 2019.
>
> [2] Stiennon N, Ouyang L, Wu J, et al. Learning to summarize with human feedback[J]. Advances in Neural Information Processing Systems, 2020, 33: 3008-3021.
>
> [3] Kreutzer J, Uyheng J, Riezler S. Reliability and learnability of human bandit feedback for sequence-to-sequence reinforcement learning[J]. arXiv preprint arXiv:1805.10627, 2018.
>
> [4] Agarwal R, Schuurmans D, Norouzi M. An optimistic perspective on offline reinforcement learning[C]//International Conference on Machine Learning. PMLR, 2020: 104-114.

---

### Official Review · Reviewer_VnTR · 2023-07-05

**Soundness:** 3 good
**Presentation:** 3 good
**Contribution:** 3 good
**Rating:** 4
**Confidence:** 3

**Summary:**

The authors propose a novel approach for training preference models (PMs) for aligning language models (LMs) that accounts for human preference diversity. Their algorithm, d-PM, trains a PM to match a distribution over human preference in human feedback data. The authors also propose a contrastive learning algorithm for aligning an LM with preferences represented by the d-PM. They validate both by conducting experiments on datasets for emotional support conversation and compliance with moral norms.

**Strengths:**

1. The paper is written clearly and was easy to follow.
2. The authors address an important, largely underexplored problem of modelling preference diversity in a principled, Bayesian fashion. The formal framework they propose is sound and elegant and I think is an interesting and original contribution to the emerging field of preference modelling for LM alignment.

**Weaknesses:**

1. The assumption that human preferences over sentences boil down to discriminating between two classes, {acceptable, unacceptable}, feels limiting. It's significantly more restrictive than the usual pairwise comparison approach in RLHF. It also makes the later claim that a distribution $l’$ over {acceptable, unacceptable} “encompass[es] all human perspectives” (line 101) implausible (e.g., it lumps together numerous ways and degrees of being unacceptable). If it’s just part of the experimental setup and not a limitation of the framework, this should me made clear, the assumption should be moved to the Experiments section and the Framework section should be more general. However, then I'm not sure the experiments will be able to provide evidence for the soundness of that more general framework.
2. Unless I’m mistaken, the method still uses point estimates of human preferences (i.e. probability of "acceptable") during the calibration phase (Step 2: Preference-based Ranking). In this sense, the framework is not fully Bayesian; to my mind, the Bayesian thing to do would to be to somehow marginalise over human preferences for each example during finetuning. I’m happy to be corrected here if I’m misunderstanding something.
3. Relatedly, I’m not sure the experimental setup explicitly tests for how effective d-PM is at making LMs find a shared consensus among disagreement. In the RoT setup, it seems that the hard work has already been done (by the annotators setting the beta values) and the LM is just trained to imitate that already-found consensus. Why couldn’t we just do that by supervised finetuning on demonstrations with high beta? (Why isn’t that a baseline?)
4. In the RoT experiment (sec 4.2.), I understand that the authors train a d-PM on the MIC dataset and then evaluate an LM aligned with d-PM against reference responses from MIC. Are these responses held-out (i.e. not used in training of d-PM)? Even if they are held-out, this setup still feels subtly wrong to me: you’re optimising the LM to imitate good MIC responses (indirectly, via d-PM scores) so it’s not surprising that calibrated models outperform base models at being similar to good MIC responses.
5. While I agree modelling preference diversity in aligning LMs is underexplored, there is some work on this and claim (i) in the Introduction (lines 62-64) feels exaggerated. See Bakker et al., 2022. Some other work that could be relevant is DPO (Rafailov et al., 2023) for calibration and Korbak et al. (2023) for a different way of framing LM alignment as Bayesian inference.
6. The authors conduct experiments with largely obsolete LMs such as BERT (2018) or T5 (2019). They could easily be replaced with, e.g., RoBERTa and FLAN-T5, respectively, which are unequivocally better in every respect and come with the same model sizes.
7. It feels a bit suspicious to evaluate d-PM’s impact on downstream LMs using a relatively obscure calibration method as opposed to more standard approaches like best-of-n sampling (Nakano et al., 2021) or finetuning on filtered data. The authors could also consider a more intrinsic evaluation, treating d-PM as a binary classifier and measuring its accuracy, F2 score or validation loss on a held-out subset of the data. It would be interesting to compare d-PM with baselines without the confounding effect of the calibration training.
8. I feel the section comparing calibration and RL finetuning is sloppy. First, it lacks essential details (algorithm being used, the exact objective, the stopping criterion); these are too important to be left for the appendix. Second, there is no reason to think that the RL loss should be comparable to a supervised learning loss. This is true in general, but especially for online RL where the objective involves chasing a moving target (on policy distribution) and is further confounded by advantage estimation and KL penalties. See [1] for an approachable explanation in the context of policy gradients [1]. To me, this is a serious red flag showing a fundamental misunderstanding on the authors’ part. If d-PM does not work well with RL, I’m quite suspicious about either authors' RL setup or d-PM itself.
10. I'd be curious to see standard errors for human evaluation scores, especially given small sample size (100).

Minor points:

10. Possible typo on line 83, “give”

11. The notation $l’ \sim l$ was somewhat unintuitive for me and $l’$ is hard to visually tell apart from $l$, at least with the current font.

[1] https://spinningup.openai.com/en/latest/spinningup/rl_intro3.html#implementing-the-simplest-policy-gradient

Bakker et al., 2022. Fine-tuning language models to find agreement among humans with diverse preferences

Korbak et al., 2022. RL with KL penalties is better viewed as Bayesian inference

Nakano et al., 2021. WebGPT: Browser-assisted question-answering with human feedback

Rafailov et al., 2023. Direct Preference Optimization: Your Language Model is Secretly a Reward Model

**Questions:**

1. How is d-PM different from training a binary classifier with soft labels {acceptable, unacceptable}, i.e. minimising the cross-entropy between model-predicted distribution over labels and empirical distribution for each example? Note that this is slightly different from the “soft” baseline in section 4.
2. In the actual implementation, is the summing over $j$ in the denominator of (4) implemented by exact enumeration of the whole dataset (for each gradient step?) or is it approximated, e.g., by sampling?
3. It was unclear to me where does the reference responses for reference-based metrics (e.g. BLEU, ROUGE) come from? Are they from the ESConv and MIC benchmarks? How these datasets split into train and test?

**Limitations:**

The authors discuss certain limitations in the final section. However, I think the certain limitations and implementation details are missing (see weaknesses 1 and 8). Moreover, experiments involve relatively small-sized models and I think the authors should be more transparent about that, e.g., report parameter counts.

I think authors' declaration `Error Bars: Yes` is misleading. There's no error bars. While statistical significance is reported for some results, not for those where I'd expect the highest variance (i.e. human evaluation with small sample size).

Finally, emotional support conversation is a sensitive domain and I'd expect the authors to highlight it a bit more and be clear that "effectiveness of the proposed method" (line 69) does not entail deployment-readiness.

---

> ### Author Rebuttal · Authors · 2023-08-10
>
> Thank you for your comprehensive feedback. We'd like to underscore the primary contribution of our work: considering the disagreement of human preference for LM alignment. Our work is focused on scenarios where human preferences are highly subjective. In such contexts, it is imperative to factor in the preferences of different people, especially since majority voting ("major") can potentially overshadow minority preferences. While a soft label ("soft") can capture this disagreement, its efficacy is sometimes compromised by outliers or extreme labels, particularly when annotations for each instance are sparse. Therefore, our approach leverages Bayesian techniques to model these human preferences, capturing the intricacies of disagreement. Concurrently, we employ Contrastive Learning (CL) to align the models with these nuanced preferences.
>
>
> $\textbf{W1:}$
>
> We acknowledge that human preferences are multifaceted and not always binary. The distinction between \{acceptable, unacceptable\} was a deliberate choice in our experimental setup to focus on scenarios where human preferences show a high degree of subjectivity. We chose this binary classification primarily because achieving broad acceptability can be very significant in scenarios with highly subjective content. This is especially true when aiming for a model's outputs to be acceptable to a wide audience rather than perfectly aligned with specific individual's nuanced preferences. Notably, line101 ("encompassing all human perspectives") indicates preferences over \{acceptable, unacceptable\} from all humans, instead of all kinds of preferences.
>
> However, we acknowledge the value of pairwise comparison approaches and the richness they can offer. Our framework, in its foundational design, is not limited to this binary setup. In the Experiments section, we will clarify that this was an experimental decision for the datasets and objectives rather than an inherent framework limitation.
>
> $\textbf{Q1,W2:}$
>
> We'd like to clarify that the Bayesian approach in our work specifically pertains to modeling human preferences via the "d-PM" method. The intention was not to incorporate a Bayesian methodology throughout the entirety of the calibration process. As highlighted in lines 13-14 and 48-49, our main focus was on capturing human preferences' inherent variability and nuances using Bayesian modeling. We believe this approach offers a more robust representation of diverse human judgments than other methods like "soft" and "major".
>
> $\textbf{W3:}$
>
> There can be some misunderstanding about the MIC dataset, and the task. The RoTs within the MIC dataset are not created based on predetermined global consensus. Instead, once a RoT is written, it is evaluated by a global consensus, which serves as an after-the-fact character for the RoT. Besides, LM is not trained to imitate the consensus, but the RoT. On the one hand, restricting our training to only high $\beta$ demonstrations would lead to reduced training data. On the other hand, demonstrations with low $\beta$ may not be the reason for misalignment.
>
> $\textbf{W4:}$
>
> While the MIC dataset is employed for training the d-PM, it's crucial to note that the d-PM is not utilized to predict the preference scores of the ground truth or for real-time generation during calibration. Moreover, during the calibration both the good and bad RoTs are used for training. This means we did not use d-PM to filter the bad RoT. Beyond the observed superior performance of Aligned$_\text{d-PM}$ compared to base models, the experiments substantiate the central hypothesis of our work. Namely, a preference model that integrates disagreement through Bayesian modeling can yield better results than models relying solely on majority voting or soft labels.
>
> $\textbf{W5:}$
>
> While we acknowledge the contributions of the mentioned papers, we'd like to clarify the distinction between our work and the cited studies. $\textbf{Bakker et al.,}$ primarily aim to generate a consensus for a specific group after assimilating their individual opinions. In contrast, our approach is directed at understanding and modeling the ``universal preference'' across different individuals, which is a broader and more encompassing goal. We believe this distinction is crucial as aligning to a group's consensus may not necessarily reflect the broader spectrum of preferences in the general populace. While $\textbf{DPO}$ provides insights into calibration, it does not delve deep into capturing and modeling the inherent disagreements in human preferences, which is the core objective of our work. $\textbf{Korbak et al.}$ offer an alternative Bayesian framing of LM alignment. However, our Bayesian approach's primary focus is to model disagreements in human preferences explicitly. The Bayesian framework in Korbak et al. is employed to solve $\textbf{the challenges of RL for LM alignment}$ and does not center on human preference disagreement. Thus, we stand by our claim that our methodology offers a novel perspective in modeling preference diversity for LM alignment.
>
> $\textbf{W9:}$
>
> The standard error is $0.74$ and $0.37$ in the two tasks, respectively. However, we argued that these values do not make sense. For tasks related to tasks involving high subjectivity, disagreement should be allowed.
>
> $\textbf{Q2:}$
>
> We appreciate the valuable feedback. In response, we conducted additional experiments. See Table 2 and overall response 2.
>
> $\textbf{Q3:}$
>
> By each gradient step.
>
> $\textbf{Q4:}$
>
> Indeed, the reference responses for metrics like BLEU and ROUGE are sourced from the ESConv and MIC datasets, including ground truths. Regarding dataset partitioning and processing, we adhere to the conventions established for the base models.
>
> $\textbf{Minor Points:}$
>
> Thank you for the advice, we change the $\mathcal{l}$ to $\rho$ to indicate the preference distribution.

---

> > ### Author Response · Authors · 2023-08-10
> > **Rebuttal for W6-8**
> >
> > $\textbf{W6:}$
> >
> > Thank you for your feedback. We acknowledge that there have been significant advancements in language models since BERT and T5. Our choice of these models was grounded in aligning with the precedent set by prior studies, ensuring comparability and continuity. While our primary objective was to demonstrate the efficacy of our proposed methodology, rather than benchmarking the latest models, we recognize the merit in using updated architectures like RoBERTa and FLAN-T5 for a more contemporary performance evaluation. Therefore, we are incorporating these newer models to validate our approach with related experiments in a more current context.
> >
> > $\textbf{W7:}$
> >
> > We appreciate this. Correspondingly, we designed corresponding experiments to validate our d-PM. However, it's important to note that d-PM isn't designed to function as a binary classifier. Instead, its outputs can be evaluated as soft labels since its primary role is to produce refined soft labels. In comparison to raw soft labels, d-PM's outputs exhibit increased robustness, particularly in mitigating the effects of outlier or extreme labels. To empirically showcase d-PM's robustness, we introduced controlled noise to our training dataset and compared the standard errors between soft labels predicted by both "soft" and d-PM. Specifically, we introduced a $20\\%$ noise level by altering the distribution across the two classes during training (loss computation). The results were telling: d-PM and "soft" yielded standard errors of $0.21$ and $0.35$ in a noise-free environment. Yet, when exposed to noise, the standard errors for d-PM remained relatively stable at $0.23$, whereas for "soft", it rose significantly to $0.47$. This empirical evidence underscores d-PM's resilience against noise, reinforcing its utility in producing reliable soft labels. We're always open to further suggestions and discussions to improve our approach and presentation.
> >
> > $\textbf{W8:}$
> >
> > There may be some misunderstanding. 1. We detailed the implementation of RL in Appendix B.5. 2. We did not compare the loss value of RL and CL directly. Figure 5 is to prove our statement in line274-276, i.e., RL yields a slow convergence speed, and CL is relatively time-efficient in this case.

---

> > ### Comment · Reviewer_VnTR · 2023-08-15
> >
> > Thanks for the detailed response and useful clarifications. However, it doesn't change my evaluation of the paper significantly: I feel the biggest weaknesses -- 4, 6, 7, 8 and 9 -- are unaddressed. Please find some follow-ups below.
> >
> > > We'd like to clarify that the Bayesian approach in our work specifically pertains to modeling human preferences via the "d-PM" method. The intention was not to incorporate a Bayesian methodology throughout the entirety of the calibration process.
> >
> > For me then, it highlights the weakness W7 (not addressed by the authors). If d-PM is the main contribution here, why was it evaluated using such a non-standard finetuning procedure?
> >
> > > The standard error is 0.74  and 0.37 in the two tasks, respectively. However, we argued that these values do not make sense. For tasks related to tasks involving high subjectivity, disagreement should be allowed.
> >
> > Wait, I'm confused what do these numbers refer to. Is it "Global Consensus" score for "Aligned d-PM" (bottom-left cell) in tables 2 and 4? Have your reported the standard error across LM outputs and annotators (for each method) or just across the three annotators? I meant the former one.
> >
> > If it's the former one, then these standard errors are quite large (rendering difference between methods insignificant under all resonant significant levels). I also don't understand how measuring standard errors does not makes sense because "disagreement should be allowed". Surely, human labellers are expected to disagree, but if the std err of their scores is much higher than difference between scores, you can't infer anything from the difference between scores.
> >
> > > While the MIC dataset is employed for training the d-PM, it's crucial to note that the d-PM is not utilized to predict the preference scores of the ground truth or for real-time generation during calibration. Moreover, during the calibration both the good and bad RoTs are used for training. This means we did not use d-PM to filter the bad RoT.
> >
> > But during calibration you are using feedback from d-PM to nudge the LM towards imitating outputs d-PM prefers and d-PM is trained to prefer good RoTs. I continue to be a bit skeptical of this evaluation if the ground truths for evaluation were not held out from training the d-PM.

---

> > > ### Comment · Reviewer_VnTR · 2023-08-18
> > >
> > > I initially missed authors' reply to W6-W8 in a separate post, sorry! Just wanted to acknowledge I've noticed them now.
> > >
> > > >  We did not compare the loss value of RL and CL directly. Figure 5 is to prove our statement in line274-276, i.e., RL yields a slow convergence speed, and CL is relatively time-efficient in this case.
> > >
> > > But convergence of loss is not necessarily tied to convergence of reward. They are not even monotonically related (because of advantage estimation, value learning, changes of on-policy distribution).

---

> > > > ### Comment · Area_Chair_j9oz · 2023-08-19
> > > > **Thanks**
> > > >
> > > > Thanks for this! It is a system failure, not your fault!

---

> > > ### Author Response · Authors · 2023-08-20
> > > **Reply to Reviewer VnTR's Concerns**
> > >
> > > Thank you for your feedback. We appreciate the opportunity to address your concerns and clarify certain points:
> > >
> > > 1. Regarding $\textbf{W2}$ \& $\textbf{W7}$: We're trying to understand if you believe W2 is related to W7. We've conducted experiments specifically for W7, showcasing that "d-P" outperforms "soft" in both prediction performance and robustness. We trust this addresses your queries about W2 and W7.
> > >
> > > 2. Regarding $\textbf{W6}$: We've incorporated additional experiments to address this. We display results for aligning MultiESC using d-PM trained based on RoBERTa in the first table; the second table represents the results of aligning Flan-T5 for the task of RoT generation.
> > >
> > > | Model | B-1 | B-2 | B-3 | B-4 | R-L | METEOR | CIDEr | Extreme |
> > > |---|:-:|:-:|:-:|:-:|:-:|:-:|:-:|:-:|
> > > | MultiESC | 20.36 | 8.80 | 4.92 | 3.14 | 21.00 | 8.58 | 30.69 | 52.74 |
> > > | Aligned$_{\text{major (RoBERTa)}}$ | 23.64 | 9.89 | 5.25 | 3.12 | 20.93 | 9.37 | 28.85 | 52.57 |
> > > | Aligned$_{\text{soft (RoBERTa)}}$ | 22.89 | 9.54 | 5.21 | 3.17 | 20.85 | 9.23 | 27.80 | 52.22 |
> > > | Aligned$_{\text{d-PM (RoBERTa)}}$ | 23.45 | 9.78 | 5.21 | 3.19 | 21.45 | 9.32 | 33.42 | 52.26 |
> > >
> > >
> > > | Decoding | Model | R-1 | R-2 | R-L | BertScore | ScareBLEU | Avg.Len |
> > > |---|:-:|:-:|:-:|:-:|:-:|:-:|:-:|
> > > |Beam | Flan-T5 | 54.99 | 34.98 | 53.65 | 93.77 | 30.70 | 8.99|
> > > |Beam | Aligned$_{\text{soft}}$ | 55.12 | 35.25 | 53.78 | 93.78 | 30.96 | 9.04 |
> > > |Beam | Aligned$_{\text{d-PM}}$ | 55.26 | 35.63 | 54.00 | 93.86 | 31.51 | 9.27 |
> > > |Greedy | Flan-T5 | 37.76 | 16.91 | 35.98 | 91.38 | 15.00 | 9.72 |
> > > |Greedy | Aligned$_{\text{soft}}$ | 37.80 | 16.78 | 35.86 | 91.37 | 14.89 | 9.75 |
> > > |Greedy | Aligned$_{\text{d-PM}}$ | 38.48 | 17.55 | 36.62 | 91.52 | 15.54 | 9.71|
> > > |$\textit{p}$=0.9 | Flan-T5 | 41.65 | 20.61 | 39.92 | 92.06 | 18.21 | 9.30 |
> > > |$\textit{p}$=0.9 | Aligned$_{\text{soft}}$ | 41.45 | 20.26 | 39.62 | 91.99 | 17.65 | 9.35 |
> > > |$\textit{p}$=0.9 | Aligned$_{\text{d-PM}}$ | 42.04 | 20.92 | 40.34 | 92.13 | 18.58 | 9.31 |
> > >
> > > We hope this can alleviate your concerns about W6.
> > >
> > > 3. Regarding $\textbf{W8}$: We concur that loss convergence alone doesn't sufficiently demonstrate model time efficiency. This is why we presented both model performance (the table in Figure 5) and loss convergence (the figure in Figure 5) to highlight the slower pace of RL training compared to CL. Notably, we prioritized performance comparison over loss convergence in both the text and figure. We'll emphasize this more explicitly in our revised manuscript. Thank you for highlighting this.
> > >
> > > 4. Regarding $\textbf{W9}$: The values $0.74$ and $0.37$ represent the standard errors among three annotators from the entire human evaluation (Table 2 \& 4) for the two tasks. This was in response to the query "$\textit{I'd be curious to see standard errors for human evaluation scores, especially given small sample}$ $\textit{size (100)}$". Based on the phrasing of the original question, it doesn't seem to pertain to "Global Consensus" or "standard error across LM outputs and annotators (for each method)". For your new question about the "standard error across LM outputs and annotators (for each method)", could you please provide more clarity?
> > >
> > > 5. Regarding the $\textbf{MIC}$ dataset ($\textbf{W4}$): If your concern revolves around us using the evaluation/testing dataset from MIC for training the d-PM, we'd like to direct your attention to lines 487-488. It states: "The split of the MIC dataset was maintained when training the model (d-PM) and calibrating the generation model to prevent data leakage."

---

> ### Author Response · Authors · 2023-08-21
> **Additional Experiments in "Author Rebuttal by Authors"**
>
> For **Q1**, we added an additional experiment in ``Author Rebuttal by Authors`` point 2: we've implemented an alternative preference model. This model considers each human label independently, without aggregation, and formulates the preference using a cross-entropy loss. We refer to this as the w/oA preference model. The performance of Aligned, when MultiESC is employed as the base model, is depicted in **Table 2**. The results indicate that its performance is somewhat diminished. A detailed analysis revealed that the preference scores generated by Aligned$_\text{w/oA}$ are relatively clustered in value in terms of value, contrasting with the wide-ranging scores yielded by d-PM. As a result, w/oA struggles to offer a logical sequencing of the generated samples.
>
> For **W8**, we kindly draw your attention to the ``Author Rebuttal by Authors`` point 3. We have added the results of our RL-based alignment (**Table 3**), which indicates the superiority of our framework to RL. We believe these results address some of the concerns raised and provide additional context to our findings.
>
> While Reinforcement Learning (RL) has proven potent in certain scenarios, as illustrated by recent research [1,2,3], a commonality among these successful applications is their training on expansive datasets (280K to 6M training instances). In contrast, our dataset's size is significantly more modest. It is possible that this small data size plays a role in the suboptimal performance of RL in our experiments. This hypothesis aligns with findings from Agarwal et al. [4], which suggest that smaller datasets can potentially undermine RL's performance.
>
> We kindly request that you review the updated RL experiment results in the rebuttal. We hope this additional information will be beneficial for the evaluation of our paper.
>
> Thank you for your time and consideration.
>
> [1] Ziegler D M, Stiennon N, Wu J, et al. Fine-tuning language models from human preferences[J]. arXiv preprint arXiv:1909.08593, 2019.
>
> [2] Stiennon N, Ouyang L, Wu J, et al. Learning to summarize with human feedback[J]. Advances in Neural Information Processing Systems, 2020, 33: 3008-3021.
>
> [3] Kreutzer J, Uyheng J, Riezler S. Reliability and learnability of human bandit feedback for sequence-to-sequence reinforcement learning[J]. arXiv preprint arXiv:1805.10627, 2018.
>
> [4] Agarwal R, Schuurmans D, Norouzi M. An optimistic perspective on offline reinforcement learning[C]//International Conference on Machine Learning. PMLR, 2020: 104-114.

---

> > ### Comment · Reviewer_VnTR · 2023-08-22
> >
> > Thanks for extensive responses. Many of my concerns were indeed addressed. I've decided to raise my score to 4. I'm still not convinced the paper meets the bar for NeurIPS which is why I'm not raising the score higher.
> >
> > > For your new question about the "standard error across LM outputs and annotators (for each method)", could you please provide more clarity?
> >
> > Yes, I meant standard error based on the variance computed by summing over both examples and annotators, so that it involves both inter-example variability and inter-annotator variability. However, I can see how one could argue against such a measure (if inter-annotator variability dominates). Inter-annotator variance is also okay. I just would like to see a claim like "A tends to be preferred over B by human annotators over ({mean_a} ± {se_a} vs {mean_b} ± {se_b})" and looks plausible given the size of the gap between A and B and compared to standard errors.

---

> > > ### Author Response · Authors · 2023-08-22
> > >
> > > Thank you very much for your detailed feedback. We truly value and appreciate the insights you've provided. Additionally, we are grateful for your consideration in adjusting our score.
> > >
> > > Besides, for **W9**, we computed the inter-annotator variances of human evaluation for different models:
> > > 1. For the task of emotional support conversation, the overall preference can be reflected by the Overall score. Its inter-annotator variances are 0.30, 0.41, 0.37, and 0.31 for Base, $Aligned_\text{major}$, $Aligned_\text{soft}$, and $Aligned_\text{d-PM}$ in Table 2, respectively.
> > > 2. For the task of RoT generation, the primary metric reflecting overall preference is the Well-formedness score. Its inter-annotator variances are 0.16, 0.17, and 0.16 for Base,  $Aligned_\text{soft}$, and $Aligned_\text{d-PM}$ in Table 4, respectively.
> > >
> > > We hope this information addresses your concerns. Thank you once again for your support and valuable comments.

---

### Official Review · Reviewer_otcY · 2023-07-06

**Soundness:** 3 good
**Presentation:** 3 good
**Contribution:** 3 good
**Rating:** 6
**Confidence:** 3

**Summary:**

The paper proposes a novel alignment method, which takes a Bayesian approach, termed Preference Modeling with Dis-
295 agreement (short as d-PM); the method is designed to capture disagreement in human labels, rather than assuming a gold human label. The Bayesian approach is basically a new way to train reward model, which will be later used to rank model generated outputs, and then apply contrastive learning loss to optimize for generating higher ranked outputs. I think the contribution mainly lies in the first part, which is to propose the new mechanism for training reward model, which are crucial for success of either RLHF or contrastive learning in this paper to align model with human preferences.

An additional contribution is the usage of contrastive learning to align Large Language Model.

**Strengths:**

The problem it aims to tackle is a very important one. Too often, AI practitioners make the possibly false assumption that human annotated labels are good, and take the naive approach of majority voting to get final training data. This approach may work when the reason of disagreement lies in annotation quality. However, it neglects the situation, which paper points out, that the disagreement distribution as a natural reflection of the diversity of opinions among populations, and should not be ignored.

The reward model training schema seems novel. I am not an expert on optimization or energy-based models; if proof in the paper does stand(which seems fine for me, but not 100% certain), it sheds light on a better method to handle annotation differences. The experiment using contrastive learning also does show some improvements, over the default approach, which is majority voting.

The experiment set-up and introduction of dataset are clear, and Table-1's result is well-organized and has most important metrics included. Human evaluation is also conducted, which further verifies the automatic evaluation. Significance tests are conducted to further verify results.

The contrastive learning setup is a combination of self-learning by generating candidates and learning reward model feedbacks. It provides another data point that contrastive learning could work, and RLHF-ppo isn't the best/only way to learn rewards.

The paper is in general easy to follow. But more backgrounds and explanation of notations will be better for Section 3.1 for non-experts to understand.

**Weaknesses:**

* For the reward model part, it's not clear to me why it's called a Bayesian approach after reading Derivation Process in 3.1; I understand that Beyesian theorem is applied to derive the final optimization loss, but isn't Beyesian theorem/ELBO are common techniques  and are also used to derive other optimization losses. I would appreciate more explanation why Beyesian is an important characteristic/contribution for this method.

* For the experiment result, looking at the Rouge-L score, the differences between majority voting and d-PM isn't huge for both Blender and MIC datasets. In some cases, like Table 4, Row-BART-Greedy, Rouge-L and other metrics are giving conflicting indications of model performance. This raises question whether the relatively modest improvements from d-PM are meaningful, when the automatic metrics are not very robust. The human evaluation does alleviate this concern, but I think it should still be noted.

* Contrastive vs RL; I think it is fine to use a contrastive learning method, rather than RL to align model; but only pointing out that "RL requires expensive online decoding procedures" does not rule out the possibility that RL could possibly give better results. Also, this paper isn't the first to use contrastive learning to align models, the results will be more convincing if an RL-baseline, and another contrastive learning baseline(maybe stable alignment  or other well-known methods) is provided.

* The evaluation scale is also limited. The experiment are only conducted on two datasets, and even on such selected domain, the improvements are modest. It is arguable how effective this approach can scale to other datasets. A wider scale evaluation on more domains will be needed.

**Questions:**

* The authors named d-PM method a Bayesian approach, but did not explicitly explain why it is the most important characteristic. If would be great if authors make that explicit in paper.

* Why only evaluate on Blender and MIC? Is there a selection criterion or hypothesis on what kinds of datasets that d-PM will work better than say majority voting?

* The paper claimed that it has expensive overheads, but I do wonder if your contrastive learning approach can have a similar effective to RL, and other robust contrastive learning methods.

**Limitations:**

Yes, addressed.

---

> ### Author Rebuttal · Authors · 2023-08-10
>
> Thank you for the detailed comments.
>
> $\textbf{W2:}$ For the experiment result, looking at the Rouge-L score, the differences between majority voting and d-PM isn't huge for both Blender and MIC datasets. In some cases, like Table 4, Row-BART-Greedy, Rouge-L and other metrics are giving conflicting indications of model performance. This raises question whether the relatively modest improvements from d-PM are meaningful, when the automatic metrics are not very robust. The human evaluation does alleviate this concern, but I think it should still be noted.
>
> To alleviate this concern, we further conducted A/B test implemented by GPT-4, shown in Table 1.
>
> $\textbf{Q1 (W1):}$ The authors named d-PM method a Bayesian approach, but did not explicitly explain why it is the most important characteristic. If would be great if authors make that explicit in paper.
>
> The Bayesian approach is the most important characteristic of our method.
> $\textit{First, as we know, this is the first work that introduces the Bayesian theorem and technologies to the field of human preferences alignment}$$\textit{for calibrating the text generation of LLMs.}$ In fact, this is not so straightforward, and an insight that why the Bayesian technology can be applied is required. Specifically, we identify the availability of such technology by diving deep into the properties of human preferences in the field of LLMs and then refer to suitable techniques in the field Bayesian to naturally address the challenge. We find that the major benefit of Bayesian technology is to allow us to refine and smooth the human preference distribution, making the refined preference potentially more accurate than the raw one, which firstly bridges the field of Bayesian technology and the field of preferences alignment for LLMs.
>
> $\textit{Second, the optimization loss is novel and elaborately designed for the field of preferences alignment based on Bayesian theories.}$
> We identify that the overall perception of the raw human annotations can be disproportionately influenced by an outlier or extreme label, potentially deviating from a "universal preference", since each text $s_i$ has a limited number of annotations. Therefore, we propose adopting Bayesian technologies to refine these annotations, aiming to approximate this "universal preference".  In this case, the observed human annotations serve as prior knowledge, and the Bayesian process leverages the distribution (likelihood probability) of all possible preference annotations to adjust and smooth the observed annotations (posterior probability). Obviously, the optimization loss is designed by incorporating the Bayesian theorem, and thus Beyesian is an important characteristic of our method. Yet, it worthwhile to note that even though our optimization loss is a application of Bayesian theorem, the design of our optimization technology requires identifying the prior knowledge and posterior knowledge, and approximating the likelihood probability in the field of human preferences alignment, which is challenging.
>
>
> $\textbf{Q2 (W4):}$ Why only evaluate on Blender and MIC? Is there a selection criterion or hypothesis on what kinds of datasets d-PM will work better than say majority voting?
>
> Our choice of the ESConv and MIC datasets was deliberate and grounded in the nature of the data they encompass. Both datasets prominently feature human sentiments, ethical norms, and other inherently subjective elements. Given this high degree of subjectivity, conventional harmonization approaches like majority voting may not be well-suited since they can potentially dilute the rich variance in human perspectives.
>
> As you rightly pointed out, this rationale is briefly touched upon in the Introduction (lines 30-32). To ensure clarity and to preclude any ambiguity, we plan to emphasize this decision-making process more explicitly at the commencement of the Experiments section.

---

> > ### Comment · Reviewer_otcY · 2023-08-17
> > **Thanks to Author Reply**
> >
> > Thank you for your rebuttal.
> >
> > As in VnTR's review(Weakness (2)), I was also having an expectation for what Bayesian method mean, but it still isn't clear to me how Bayesian is your described approach after your explanation.
> >
> > I can agree with authors' comment that the optimization loss is a novel point, and the observation and accommodation for outliers.
> >
> > Still, the weaknesses mentioned above mostly remains, I will not change the current recommendation.

---

> > > ### Author Response · Authors · 2023-08-21
> > > **Explanation for Bayesian method**
> > >
> > > Thank you very much for your response.
> > >
> > > **The core idea of our Bayesian method is to calibrate the human annotation preferences for each text. ** As we stated in the paper, human preferences are usually inconsistent among texts.
> > > - For example, given a text $x$, three annotation humans may give various preferences (can be viewed as labels in classification tasks), i.e., $y_1, y_2, y_3$, where $y_1=y_2 \neq y_3$. To train the reward model, one may utilize a majority vote to decide the final label, i.e., endow the text $x$ with label $y_1$. However, the final label may not be correct when introducing more annotation humans, e.g., $y_3=y_4=y_5$ when there 5 humans. Intuitively, the final preference of a text is the most general when all annotators give their preferences for this text. But, the human cost is too high to be practical. Hence, it is necessary to refer to the annotation information from other texts to calibrate the annotation of the given text.  However, utilizing information from other texts poses a challenge. Therefore, we have adopted the Bayesian approach to address this issue.
> > >
> > > - Practically, each text is usually annotated by only one human [1]. Specifically, each text $x$ is only endowed with a single annotation $y$ by a human. This annotation may include a specific preference of this annotator. While the majority vote-based method cannot calibrate the annotation in this case. To solve this problem, an intuitive idea is to consider another two neighbors $(x_1', y_1')$ and $(x_2', y_2')$ of this text $x$, assuming that they have the same ground-truth preference. Then, one can utilize the majority note on three labels of $y_1', y_2', y$ to decide the final correct annotation. However, how to judge the consistency between neighbors is challenging, and the number of labels is still limited even using neighbors.
> > >
> > > Considering this, we propose utilizing the Bayesian approach to learn the distribution of all texts and incorporate their information to calibrate the annotation for each text. To some extent, our method can be viewed as implicitly allowing all humans to annotate each text and then applying a majority vote on them. Due to the large number of annotators for each text, the preference of each text is consequently more general.
> > >
> > > We believe that the criterion for evaluating the characteristics of a method lies not in whether it constitutes a new optimization loss, but rather in how this loss is derived. In the end, many training methods fundamentally involve creating a new loss function as their core approach. For example, Meta Learning is to design a novel bi-level optimization loss; Contrastive Learning is to design a novel contrastive Loss; Knowledge Distillation is to construct a novel loss by using the soft label.  Similar to these methods, the loss of our method is designed mainly through the Bayesian technology, and thus it is the main characteristic of our method.
> > >
> > > [1] Training language models to follow instructions with human feedback

---

> ### Author Response · Authors · 2023-08-21
> **Additional RL Experiments in "Author Rebuttal by Authors"**
>
> We would like to kindly draw your attention to the ``Author Rebuttal by Authors`` section, specifically the third point. In that section, we have added the results of our RL-based alignment (**Table 3**), which indicates the superiority of our framework to RL. We believe these results address some of the concerns raised and provide additional context to our findings.
>
> While Reinforcement Learning (RL) has proven potent in certain scenarios, as illustrated by recent research [1,2,3], a commonality among these successful applications is their training on expansive datasets (280K to 6M training instances). In contrast, our dataset's size is significantly more modest. It is possible that this small data size plays a role in the suboptimal performance of RL in our experiments. This hypothesis aligns with findings from Agarwal et al. [4], which suggest that smaller datasets can potentially undermine RL's performance.
>
> We kindly request that you review the updated RL experiment results in the rebuttal. We hope this additional information can alleviate your concerns about  **W3** and will be beneficial for the evaluation of our paper.
>
> Thank you for your time and consideration.
>
> [1] Ziegler D M, Stiennon N, Wu J, et al. Fine-tuning language models from human preferences[J]. arXiv preprint arXiv:1909.08593, 2019.
>
> [2] Stiennon N, Ouyang L, Wu J, et al. Learning to summarize with human feedback[J]. Advances in Neural Information Processing Systems, 2020, 33: 3008-3021.
>
> [3] Kreutzer J, Uyheng J, Riezler S. Reliability and learnability of human bandit feedback for sequence-to-sequence reinforcement learning[J]. arXiv preprint arXiv:1805.10627, 2018.
>
> [4] Agarwal R, Schuurmans D, Norouzi M. An optimistic perspective on offline reinforcement learning[C]//International Conference on Machine Learning. PMLR, 2020: 104-114.

---

### Official Review · Reviewer_Ru2e · 2023-07-07

**Soundness:** 3 good
**Presentation:** 3 good
**Contribution:** 3 good
**Rating:** 6
**Confidence:** 4

**Summary:**

This paper proposed to train a reward model to capture the disagreement among human raters and use it to calibrate NLG model's generation.  The contributions includes: align text generation modes with human preference considering disagreement, a d-PM (preference modeling with disagreement) approach to train reward model, experiments on two NLG tasks of emotional support conversations and integrity ROT generation.

**Strengths:**

- exploring and emphasizing the problem of human disagreement in preference modeling
- proposed a bayesian approach to better modeling the human disagreement
- experiments on two NLG tasks with human evaluation that shows the gain of the proposed method

**Weaknesses:**

- a simple baseline for reward model should be included: taking each human label as independent (without aggregation), with a cross entropy loss.

- auto eval metrics for the 2 tasks are all reference based metrics, these metrics doesn't fully taken account into how well the model respects human preferences. Instead a reward model based auto eval metric should be used instead. Reference can also be evaluated using these reward model based metrics which also tells you how good references are. However the human evals capture that, but there lacks data on statistical significancy.

- one main claim of the paper is "such methods suffer from an inability to capture the nuanced degrees of disaggregation among humans and may only represent a specialized subset of individuals, thereby lacking the ability to quantitatively disclose the universality of human preferences." There is no direct experimental results to measure or support this. I would suggest to weaken this argument if no direct evidences can be provided.

**Questions:**

- it is a bit difficult to understand Figure2.a , more training details of the model (e.g. an algorithm description) could be added in the main text to help reader understand. Is my understanding correct that the prior is the "universal preference" estimated by the R(si, ci, \theta) and new reward is the observed preference multiplies by the prior?

- "harmonizing preference disagreements can inadvertently disadvantage minority groups" is there any reference to this? why is the bayesian approach with a "universal preference" different? Do you have any experimental results to support it?

- (minor suggestion) in many cases "preference" data refers to model generating two sequences are collecting data directly on which one is better. in this paper, it is a binary classification on each individual sequence of good/bad. "preference modeling" is a little bit misleading here. might be good to use a different word or clarify this clearly.

---

> ### Author Rebuttal · Authors · 2023-08-10
>
> We are extremely grateful for the positive assessment and highly detailed and constructive feedback on our paper.
>
> $\textbf{W1:}$ a simple baseline for reward model should be included: taking each human label as independent (without aggregation), with a cross entropy loss.
>
> We appreciate the valuable feedback. In response, we conducted additional experiments using MultiESC as our dataset. The findings are presented in Table 2 and underscore the superior performance of our proposed model Aligned$_\text{d-PM}$.
>
> $\textbf{W2:}$ auto eval metrics for the 2 tasks are all reference based metrics, these metrics doesn't fully taken account into how well the model respects human preferences. Instead a reward model based auto eval metric should be used instead. Reference can also be evaluated using these reward model based metrics which also tells you how good references are. However the human evals capture that, but there lacks data on statistical significancy.
>
> We conducted an A/B test utilizing GPT-4, the results of which are displayed in Table 1.
>
>
> $\textbf{W3:}$ one main claim of the paper is "such methods suffer from an inability to capture the nuanced degrees of disaggregation among humans and may only represent a specialized subset of individuals, thereby lacking the ability to quantitatively disclose the universality of human preferences." There is no direct experimental results to measure or support this. I would suggest to weaken this argument if no direct evidences can be provided.
>
> Thank you very much for the constructive suggestion. We in this paper implemented two other preference models: "major" and "soft". Through subsequent text generation experiments using these models, we show that the texts generated by these methods achieve fewer preferences of individuals than our method, which partially verifies this argument. Yet, we agree that we have no direct results to measure the specialized set of individuals and support this statement theoretically from a general case. Hence, we refine our argument as: "such methods might have limitations in capturing the nuanced degrees of disaggregation among humans, potentially focusing on a small subset of individuals, and thus might not fully reveal the universality of human preferences."
>
>
> $\textbf{Q1:}$ It is a bit difficult to understand Figure2.a, more training details of the model (e.g. an algorithm description) could be added in the main text to help reader understand. Is my understanding correct that the prior is the "universal preference" estimated by the $\mathcal{R}(s,c;\theta)$ and new reward is the observed preference multiplies by the prior?
>
> Thank you for this advice, and we will refine the main text in Section3.1 to align with the Figure2.a.
> Your understanding is on the right track, but it's essential to clarify that the variables we use don't correspond to "rewards". Instead, they represent posteriors within a generative model of $s$$\textemdash$ denoted as $p(s|c_i,\mathcal{l})$. Specifically, $r_i(\mathcal{l})$ refers to the $\textbf{true posterior}$, while the variable "universal preference" as estimated by $\mathcal{R}(s,c;\theta)$ serves as an $\textbf{estimated posterior}$.
>
> In Section3.1, we propose to infer the "universal preference" by the optimization of the generative model\textemdash minimization of the $\textbf{free energy}$, as outlined in Equation (2). The content after Equation (2) in Section3.1 is to derive the minimization.
> minority
>
> $\textbf{Q2:}$ "harmonizing preference disagreements can inadvertently disadvantage minority groups" is there any reference to this? why is the bayesian approach with a "universal preference" different? Do you have any experimental results to support it?
>
> Yes. Harmonization strategies such as majority voting can lead to overrepresenting majority preferences, potentially sidelining minority viewpoints [2,12] (references in the paper). This can be particularly problematic in situations where diverse perspectives are essential, like the situations in our paper. Our proposed Bayesian approach, "d-PM", offers a nuanced difference. Instead of converging to a single majority perspective, "d-PM" captures the distribution of all human preferences by refining the annotation of a text with its probability (Eq.(4) and Eq.(5)), ensuring that each annotator's preference is taken into account. This is distinct from simple aggregation methods like majority voting. To empirically demonstrate the efficacy of our approach, we implemented another preference model "major", which predominately models majority voting annotations. The results clearly indicated that Aligned$\text{d-PM}$ consistently outperformed Aligned$_\text{major}$.
>
> $\textbf{Q3}$
>
> Thank you for this advice. We decide to clarify the preference modeling in the introduction.

---

> > ### Comment · Area_Chair_j9oz · 2023-08-18
> > **Please Reply to Author Rebuttal**
> >
> > Dear reviewer,
> >
> > Thanks a lot for your efforts and valuable reviews. Would you please check this author rebuttal and see how they address your concerns on baseline, metrics, and unsupported claim? Please reply to authors by adding your following comments below this author rebuttal.
> >
> > As the author-reviewer discussion is closed soon, we would appreciate if you could submit your reply to authors by Aug 21st 1pm EDT.
> >
> > Thanks!
> >
> > Best,
> > AC

---

### Author Rebuttal · Authors · 2023-08-10

We thank all the reviewers for their comments and valuable feedback. We have made the following major updates to improve our work further:

1. Following the suggestion of Reviewer $\color{blue}{\textbf{Ru2e}}$, we adopt GPT-4 as a reward model-based automatic evaluation metric to evaluate models. Results are shown in Table 1. We use GPT-4 to compare two outputs generated by two different models given the same context, and select the more preferred one. For each pair of generations, we change their position and compare twice with the temperature of GPT-4 as $1$. After the first two comparisons, there are $66.67\%$ comparison pairs that achieve a consistent result. For the inconsistent pairs, we set the temperature of GPT-4 as $0.5$ and compare them again. We also compare the results between GPT-4 and human annotations on the $200$ samples selected for human evaluation, and their inter-rater agreement (Fleiss' kappa) is 0.43, indicating a moderate agreement.

2. Based on the feedback from $\color{blue}{\textbf{Ru2e}}$ and $\color{red}{\textbf{VnTR}}$, we've implemented an alternative preference model. This model considers each human label independently, without aggregation, and formulates the preference using a cross-entropy loss.  We refer to this as the w/oA preference model. The performance of Aligned$_\text{w/oA}$, when MultiESC is employed as the base model, is depicted in Table 2. The results indicate that its performance is somewhat diminished. A detailed analysis revealed that the preference scores generated by w/oA are relatively clustered in value in terms of value, contrasting with the wide-ranging scores yielded by d-PM. As a result, w/oA struggles to offer a logical sequencing of the generated samples.

3. We're grateful for the feedback from Reviewer $\color{pink}{\textbf{otcY}}$, $\color{red}{\textbf{VnTR}}$, and $\color{green}{\textbf{mWz9}}$. In response, we have incorporated an RL-based model; the corresponding results can be found in Table 3.

While Reinforcement Learning (RL) has proven potent in certain scenarios, as illustrated by recent research [1,2,3], a commonality among these successful applications is their training on expansive datasets (280K to 6M training instances). In contrast, our dataset's size is significantly more modest. It is possible that this small data size plays a role in the suboptimal performance of RL in our experiments. This hypothesis aligns with findings from Agarwal et al. [4], which suggest that smaller datasets can potentially undermine RL's performance.

If the manuscript is accepted, all the above content will be merged into the main text, given the extra page limit for the camera-ready version.

[1] Ziegler D M, Stiennon N, Wu J, et al. Fine-tuning language models from human preferences[J]. arXiv preprint arXiv:1909.08593, 2019.

[2] Stiennon N, Ouyang L, Wu J, et al. Learning to summarize with human feedback[J]. Advances in Neural Information Processing Systems, 2020, 33: 3008-3021.

[3] Kreutzer J, Uyheng J, Riezler S. Reliability and learnability of human bandit feedback for sequence-to-sequence reinforcement learning[J]. arXiv preprint arXiv:1805.10627, 2018.

[4] Agarwal R, Schuurmans D, Norouzi M. An optimistic perspective on offline reinforcement learning[C]//International Conference on Machine Learning. PMLR, 2020: 104-114.

---

### Decision · Program_Chairs · 2023-09-21

**Decision:**

Accept (poster)

**Comment:**

This paper proposes to incorporate disagreements of human preferences to train the reward model. To this end, it presents a Bayesian framework to model the distribution of human preferences to train a reward model, and then use contrastive learning to align the language models with the reward model. Reviewers appreciate that this is a novel alignment approach to address disagreement in human preferences, which is an important yet under-explored problem. To address reviewers' concerns, during rebuttal, authors further provided two crucial baselines of (1) considering human labels independently without aggregation, and (2) training with Reinforcement Learning. Such comparisons further back up the effectiveness of the proposed method. In addition to the new results, the authors are encouraged to improve writing to address reviewers's concerns over the details of the Bayesian approach.